# Single-exposure visual memory judgments are reflected in inferotemporal cortex

Travis Meyer, Nicole C Rust*

Department of Psychology, University of Pennsylvania, Philadelphia, United States

**Abstract** Our visual memory percepts of whether we have encountered specific objects or scenes before are hypothesized to manifest as decrements in neural responses in inferotemporal cortex (IT) with stimulus repetition. To evaluate this proposal, we recorded IT neural responses as two monkeys performed a single-exposure visual memory task designed to measure the rates of forgetting with time. We found that a weighted linear read-out of IT was a better predictor of the monkeys' forgetting rates and reaction time patterns than a strict instantiation of the repetition suppression hypothesis, expressed as a total spike count scheme. Behavioral predictions could be attributed to visual memory signals that were reflected as repetition suppression and were intermingled with visual selectivity, but only when combined across the most sensitive neurons.
DOI: https://doi.org/10.7554/eLife.32259.001

## Introduction

The everyday act of viewing the things around us leaves us with memories of the things that we have encountered. Under the right conditions, this type of 'visual recognition memory' can be quite remarkable. For example, after viewing thousands of images, each only once and only for a few seconds, we can determine with high accuracy the specific images that we have viewed (*Brady et al., 2008*; *Standing, 1973*). Additionally, we can remember not just the objects that we've seen, but also the specific configurations and contexts we saw them in (*Brady et al., 2008*), suggesting that our brains store these memories with considerable visual detail. Where and how are visual memories stored and where and how is the percept of visual memory signaled?

One candidate mechanism for signaling visual memory percepts is the adaptation-like response reduction that occurs in high-level visual brain areas with stimulus repetition, known as 'repetition suppression' (*Fahy et al., 1993*; *Li et al., 1993*; *Miller and Desimone, 1994*; *Riches et al., 1991*; *Xiang and Brown, 1998*). Consistent with that proposal, individual viewings of a novel image produce response reductions in inferotemporal cortex (IT) that can last tens of minutes to hours (*Fahy et al., 1993*; *Xiang and Brown, 1998*). Signaling visual memories in this way is attractive from a computational perspective, as it could explain how IT supports visual identity and visual memory representations within the same network. That is, insofar as visual representations of different images are reflected as distinct patterns of spikes across the IT population (*Figure 1*; *DiCarlo et al., 2012*; *Hung et al., 2005*), this translates into a population representation in which visual information is reflected by the population vector angle (*Figure 1*). If it were the case that visual recognition memories were reflected by changes in the total numbers of spikes or equivalently population response vector length, this could minimize interference when superimposing visual memories and visual identity representations within the same network (*Figure 1*).

While attractive, there are also reasons to question whether visual memory percepts manifest purely as repetition suppression in IT cortex. For example, following many repeated image exposures (e.g. hundreds to thousands), IT neurons exhibit tuning sharpening (*Anderson et al., 2008*; *Freedman et al., 2006*), and a subset of neurons reflect tuning peak enhancement (*Lim et al., 2015*; *Woloszyn and Sheinberg, 2012*), and these changes could happen during single-exposure memory

*For correspondence:
nrust@sas.upenn.edu

**eLife digest** As we go about our daily lives, we store visual memories of the objects and scenes that we encounter. This type of memory, known as visual recognition memory, can be remarkably powerful. Imagine viewing thousands of images for only a few seconds each, for example. Several days later, you will still be able to distinguish most of those images from previously unseen ones. How does the brain do this?

Visual information travels from the eyes to an area of the brain called visual cortex. Neurons in a region of visual cortex called inferotemporal cortex fire in a particular pattern to reflect what is being seen. These neurons also reflect memories of whether those things have been seen before, by firing more when things are new and less when they are viewed again. This decrease in firing, known as repetition suppression, may be the signal in the brain responsible for the sense of remembering.

Meyer and Rust have now tested this idea by training macaque monkeys to report whether images on a screen were new or familiar. The monkeys were very good at remembering the images they had seen more recently, although they tended to forget some of the images with time. Then, the rate at which the monkeys forgot the images was compared with the rate at which repetition suppression disappeared in inferotemporal cortex. The results showed that the total number of firing events in this region was not a great predictor of how long the monkeys remembered images. However, a decrease in the number of firing events for a particular subset of the neurons did predict the remembering and forgetting. Repetition suppression in certain inferotemporal cortex neurons can thus account for visual recognition memory.

Brain disorders and aging can both give rise to memory deficits. Identifying the mechanisms underlying memory may lead to new treatments for memory-related disorders. Visual recognition memory may be a good place to start because of our existing knowledge of how the brain processes visual information. Understanding visual recognition memory could help us understand the mechanisms of memory more broadly.

DOI: https://doi.org/10.7554/eLife.32259.002

as well. Similarly, in the case of highly familiar images, neurons in a brain area that lie beyond IT, perirhinal cortex, are reported to signal familiarity with increases (as opposed to decreases) in firing rate (*Tamura et al., 2017*) and highly familiar faces produce larger perirhinal fMRI BOLD responses as compared to faces that are unfamiliar (*Landi and Freiwald, 2017*). In humans, tests of the hypothesis that limited-exposure visual memory percepts are supported by repetition suppression signals have produced mixed results, with some studies providing support (*Gonsalves et al., 2005*; *Turk-Browne et al., 2006*) and others refuting the hypothesis (*Ward et al., 2013*; *Xue et al., 2011*). Additionally, studies have implicated factors beyond overall response strength in limited-exposure familiarity, including the repeatability of human fMRI response patterns across exposures (*LaRocque et al., 2013*; *Xue et al., 2010*) and synchronization between gamma band oscillations and spikes in monkey hippocampus (*Jutras et al., 2013*). Notably, while a number of studies have investigated limited-exposure repetition suppression effects in IT at the resolution of individual-units (*De Baene and Vogels, 2010*; *Li et al., 1993*; *McMahon and Olson, 2007*; *Ringo, 1996*; *Sawamura et al., 2006*; *Verhoef et al., 2008*; *Xiang and Brown, 1998*), no study to date has attempted to determine whether these putative visual memory signals can in fact account for visual memory behaviors.

To evaluate the hypothesis that repetition suppression in IT accounts for familiarity judgments during a visual memory task, we trained two monkeys to view images and report whether they were novel (had never been seen before) or were familiar (had been seen exactly once), across a range of delays between novel and familiar presentations. To explore the IT representation of visual memory on both correct and error trials, we parameterized the task such that visual memories were remembered over a timescale of minutes within experimental sessions that lasted approximately one hour. We found that while both monkeys displayed characteristic forgetting functions and reaction time patterns, these behavioral patterns were not well-predicted by a spike count decoder that embodied the strictest interpretation of the repetition suppression hypothesis. These behavioral patterns were

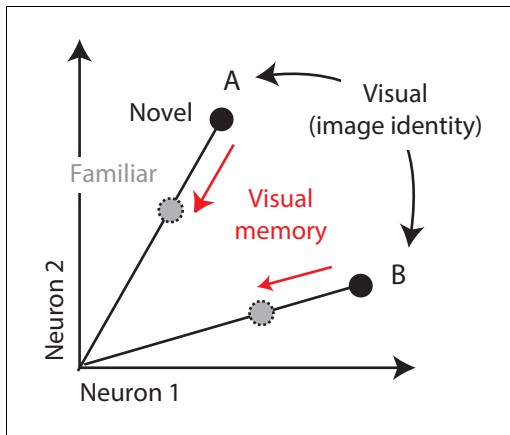

**Figure 1.** Multiplexing visual and visual memory representations. Shown are the hypothetical population responses to two images (**A** and **B**), each presented as both novel and familiar, plotted as the spike count response of neuron 1 versus neuron 2. In this scenario, visual information (e.g. image or object identity) is reflected by the population response pattern, or equivalently, the angle that each population response vector points. In contrast, visual memory information is reflected by changes in population vector length (e.g. a multiplicative rescaling with stimulus repetition). Because visual memory does not impact vector angle in this hypothetical scenario, superimposing visual memories in this way would mitigate the impact of single-exposure plasticity on the underlying perceptual representation.

DOI: https://doi.org/10.7554/eLife.32259.003

better accounted for by a linear read-out that weighted each neuron proportional to the amount of visual memory information reflected in its responses.

## Results

### The single-exposure visual memory task

While compelling, the robustness with which visual memories are stored also presents a challenge to investigating their underlying neural correlates. Ideally, investigations of the neural signals supporting a behavior are made in a context where a task is parametrically varied from easy-to-challenging, and one can evaluate the degree to which behavioral sensitivities and behavioral confusions are reflected in neural responses (*Parker and Newsome, 1998*). Following on visual recognition memory studies that demonstrate a relationship between the time that images are viewed and how well they are remembered (*Brady et al., 2009*; *Potter and Levy, 1969*), we increased task difficulty by reducing image viewing time from the 2–3 s used in the classic human visual recognition memory studies to 400 ms. To titrate task difficulty within this regime, we explored a range of delays between novel and repeated presentations.

In these experiments, two monkeys performed a task in which they viewed images and indicated whether they were novel or familiar with an eye movement response. Monkeys initiated each trial by fixating a point at the center of the screen, and this was followed by a brief delay and then the presentation of an image (*Figure 2a*). After 400 ms of fixating the image, a go cue appeared, indicating that the monkeys were free to make their selection via a saccade to one of two response targets (*Figure 2a*). Correct responses were rewarded with juice. While the first image presented in each session was always novel, the probability of subsequent images being novel versus familiar quickly converged to 50%. Novel images were defined as those that the monkeys had never viewed before (in the entire history of training and testing) whereas familiar images were those that had been presented only once, and earlier in the same session. A representative set of images can be found in *Figure 2—figure supplement 1*. Delays between novel and familiar presentations (*Figure 2b*) were pseudorandomly selected from a uniform distribution, in powers of two (n-back = 1, 2, 4, 8, 16, 32 and 64 trials corresponding to mean delays of 4.5 s, 9 s, 18 s, 36 s, 1.2 min, 2.4 min, and 4.8 min, respectively). To prevent confusion, we emphasize that our usage of the term 'n-back' refers to the numbers of trials between novel and familiar presentations, in contrast to the usage of this term in other studies that required a same/different comparison between the current stimulus and a stimulus presented a fixed number of trials back (e.g. 2-back) in a block design (e.g. *Cornette et al., 2001*).

The monkeys' performance on this task was systematic, as illustrated by smoothly declining 'forgetting functions', plotted as the proportion of trials that images were reported familiar as a function of n-back (i.e. the number of trials between novel and familiar presentations; *Figure 3a,c*). When familiar images were immediately repeated (n-back = 1), both monkeys most often called them familiar (proportion chose familiar = 0.98 and 0.94; *Figure 3a,c*). Similarly, when images were novel, monkeys were unlikely to call them familiar (proportion chose familiar = 0.13 and 0.07; *Figure 3a,c*).

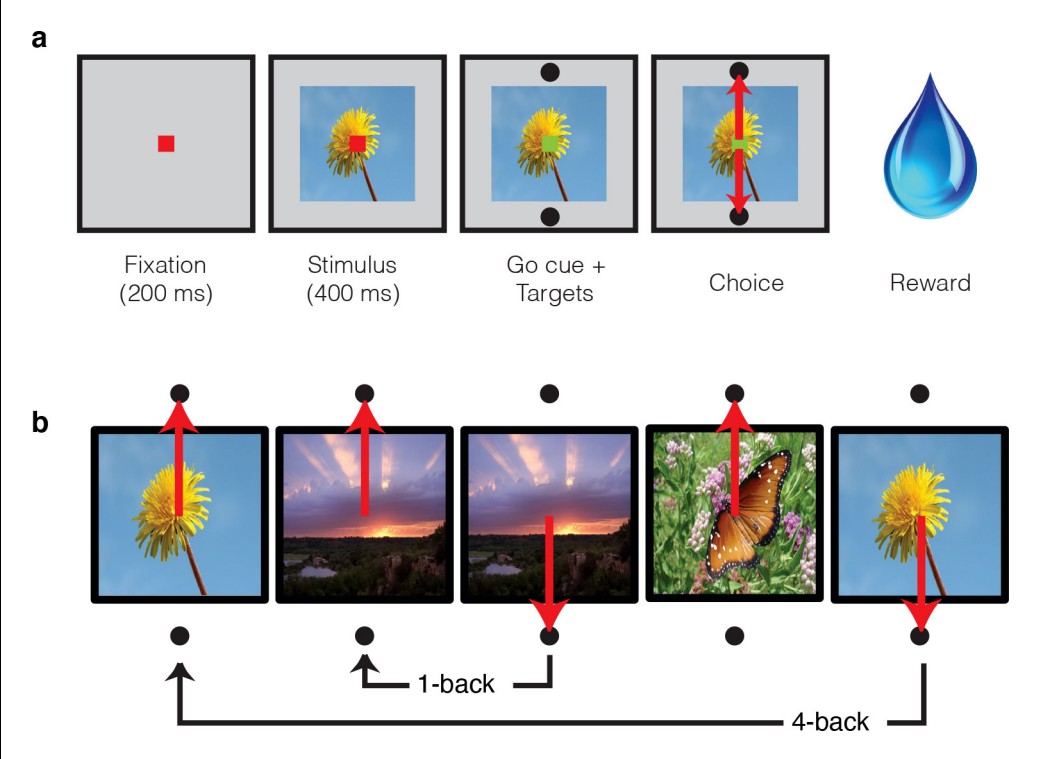

**Figure 2.** Single-exposure visual memory task. In this task, monkeys viewed images and reported whether they were novel (i.e. had never been encountered before) or were familiar (had been encountered once and earlier in the same session) across a range of delays between novel and repeated presentations. (**a**) Each trial began with the monkey fixating for 200 ms. A stimulus was then shown for 400 ms, followed by a go cue, reflected by a change in the color of the fixation dot. Targets, located above and below the image, were associated with novel and familiar selections, and differed for each monkey. The image remained on the screen until a fixation break was detected. Successful completion of the trial resulted in a juice reward. (**b**) Example sequence where the upward target was associated with novel images, and the downward target with familiar images. Familiar images were presented with n-back of 1, 2, 4, 8, 16, 32, and 64 trials, corresponding to average delays of 4.5 s, 9 s, 18 s, 36 s, 1.2 min, 2.4 min, and 4.8 min, respectively. Additional representative images can be found in *Figure 2—figure supplement 1*.

DOI: https://doi.org/10.7554/eLife.32259.004

The following figure supplement is available for figure 2:

**Figure supplement 1.** Representative images used in the experiment, sampled from http://commons.wikimedia. org/wiki/Main_Page under the Creative Commons Attribution 4.0 International Public License https:// creativecommons.org/licenses/by/4.0/.

DOI: https://doi.org/10.7554/eLife.32259.005

Between these two extremes, the proportion of familiar reports systematically decreased as a function of n-back (*Figure 3a,c*). In monkey 1, performance at 32 and 64 back fell below chance (32-back = 0.46, 64-back = 0.27, chance = 0.50), indicating that this animal most often reported that familiar images repeated after these longer delays were novel (*Figure 3a*). In monkey 2, performance at 32 and 64 back remained above chance (32-back = 0.76, 64-back = 0.54), indicating higher performance in this animal as compared to monkey 1 (*Figure 3c*).

We also analyzed reaction times for novel and familiar trials, parsed by correct and error trial outcomes. Reaction times were measured relative to the onset of the go cue (which appeared 400 ms after stimulus onset). We found that mean reaction times on correctly reported familiar trials systematically increased as a function of n-back, or equivalently, reaction times on correct trials increased with increasing difficulty (*Figure 3b,d* red). Conversely, reaction times on error trials decreased as a function of n-back, or equivalently, reaction times on error trials decreased with increasing difficulty (*Figure 3b,d*, cyan). In both animals, this led to an x-shaped pattern in the mean reaction times on

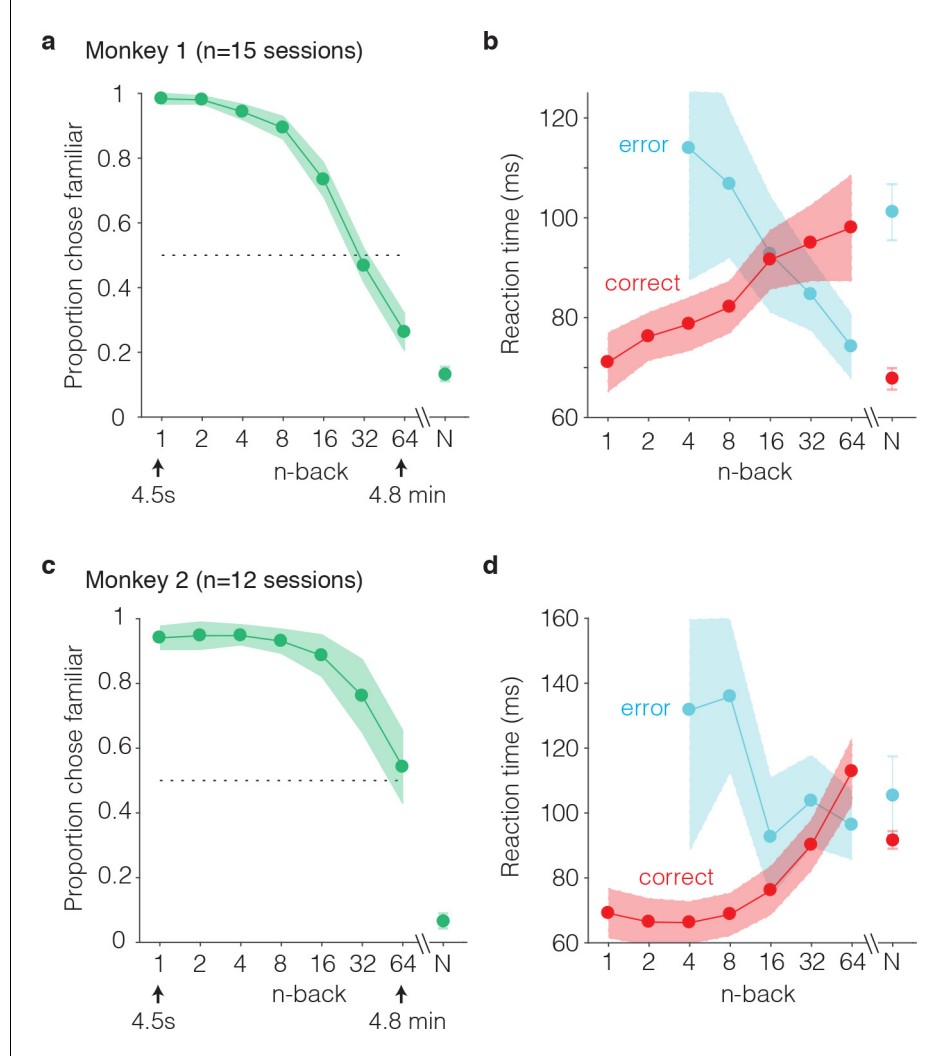

**Figure 3.** Behavioral performance of two monkeys on the single-trial visual recognition memory task. (a,c) 'Forgetting functions', plotted as the proportion of trials that each monkey reported images as familiar as a function of the number of trials between novel and repeated presentations (n-back). Novel trials are indicated by 'N' and a break in the x-axis. The dotted line indicates chance performance on this task, 50%. Error bars depict 97.5% confidence intervals of the per-session means. (b,d) Mean reaction times, parsed according to trials in which the monkeys answered correctly versus made errors. Reaction times were measured relative to onset of the go cue, which was presented at 400 ms following stimulus onset. Error bars depict 97.5% confidence intervals computed across all trials.

DOI: https://doi.org/10.7554/eLife.32259.006

familiar trials when plotted as a function of n-back. On novel trials, reaction times mimicked the pattern observed for the low n-back familiar cases in that reaction times were faster on correct as compared to error trials (*Figure 3b,d*).

From what underlying process might these x-shaped reaction time patterns arise? As is the case for nearly any task, behavioral performance can be thought of as the outcome of passing a signal (in this case a memory signal) through a decision process. The x-shaped patterns that we observed differ from the patterns reported for tasks that are well-accounted for by the standard drift diffusion model (DDM) of decision making, such as the dot-motion-direction task (*Gold and Shadlen, 2007*). In agreement with standard DDM predictions, reaction times on correct trials increased as task performance decreased (i.e. with n-back). However, reaction times on error trials decreased with n-back whereas the standard DDM predicts that reaction times will be matched on correct and error trials

(and thus reaction times on error trials should increase with n-back as well). While it is the case that extensions to this framework can predict reaction time asymmetries (*Ratcliff and McKoon, 2008*), additional parameters are required for it to do so, and these additions make it less well-suited for the purposes of this study (focused on evaluating the plausibility that IT visual memory signals can quantitatively account for visual memory behavior). We have, however, determined that these x-shaped reaction time patterns can be captured by a very simple, low-parameter extension to the signal detection theory framework, as proposed by 'strength theory' (*Murdock, 1985*; *Norman and Wickelgren, 1969*). Like signal detection theory, strength theory proposes that a noisy internal variable ('memory strength') is compared to a criterion to determine whether an image is novel or familiar (*Figure 4*, left). Strength theory also predicts that during a visual memory task, reaction times will be inversely related to the distance of this variable from the criterion, loosely analogous to a process in which increased certainty produces faster responses (*Figure 4*, middle). This leads to the qualitative prediction that when images are repeated with short n-back, memories are strong, and this will produce reaction times that are faster on correct as compared to error trials (*Figure 4*, red vs. blue). In contrast, at long n-back, memories are weak, and this will produce reaction times that are slower on correct as compared to error trials (*Figure 4*, green vs. purple). The combined consequence of strong and weak memories is an x-shaped pattern.

In sum, the reproducible patterns reflected in both monkeys' forgetting functions, along with their reaction time patterns, place non-trivial constraints on the candidate neural signals that account for single-exposure visual memory behavior. The x-shaped patterns of reaction times that we observe cannot be accounted for by a standard drift diffusion process, but they can, in principle, be captured by the predictions of strength theory. However, a successful description of the neural signals supporting single-exposure visual memory behavior requires identifying a neural signal whose sensitivity to the elapsed time between initial and repeated presentations of an image matches the sensitivity reflected in the monkeys' behavior.

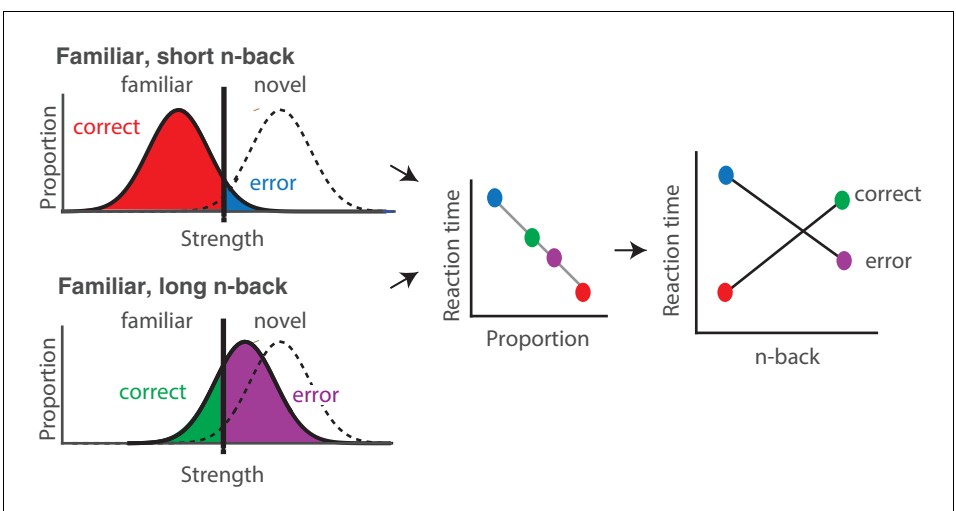

**Figure 4.** Strength theory qualitatively predicts x-shaped reaction time patterns. Like signal detection theory, strength theory proposes that the value of a noisy internal variable, memory strength, is compared to a criterion to differentiate novel versus familiar predictions. *Left:* shown are the hypothetical distributions of memory strengths across a set of images presented as novel (dashed lines) and as familiar (black), repeated after a short (top) versus long (bottom) delay. The colored portions of each familiar distribution indicate the proportion of trials corresponding to correct reports and errors, based on the position of the distribution relative to the criterion. In the case of short n-back, memory strength is high, the proportion correct is high, and the proportion wrong is low. In the case of long n-back, memory strength is low, the proportion correct is low and the proportion wrong is high. *Middle:* strength theory proposes an inverted relationship between proportion and mean reaction times, depicted here as linear. *Right:* passing the distributions on the left through the linear function in the middle produces an x-shaped reaction time pattern.
DOI: https://doi.org/10.7554/eLife.32259.007

## Single-exposure visual memory signals in IT cortex

As monkeys performed this task, we recorded neural responses from IT using multi-channel probes acutely lowered before the beginning of each session. For quality control, recording sessions were screened based on their neural recording stability across the session, their numbers of visually responsive units, and the numbers of behavioral trials completed (see Materials and methods). The resulting data set included 15 sessions for monkey 1 (n = 403 units), and 12 sessions for monkey 2 (n = 396 units). Both monkeys performed many hundreds of trials during each session (~600–1000, corresponding to ~300–500 images each repeated twice). The data reported here correspond to the subset of images for which the monkeys' behavioral reports were recorded for both novel and familiar presentations (e.g. trials in which the monkeys did not prematurely break fixation during either the novel or the familiar presentation of an image).

We began by considering the proposal that the signals differentiating novel versus familiar presentations of images were systematically reflected as response decrements with stimulus repetition (i.e. 'repetition suppression'). As a first, simple illustration of the strength of these putative single-exposure memory signals, shown in *Figure 5a* is a plot of the grand mean firing rate response of all 799 units parsed by n-back, plotted as a function of time relative to stimulus onset. This plot reveals a fairly systematic decrement in the response with repetition that diminished with time since the novel presentation. We quantified the magnitude of suppression as the decrement in the area under each n-back trace relative to the novel trace, computed 150–400 ms after stimulus onset (*Figure 5b*). Consistent with a visual memory signal that degrades (or forgets) with time, immediate stimulus repetition resulted in a decrement in the response of ~11% and suppression magnitudes decreased systematically with n-back. Also, qualitatively consistent with the repetition suppression hypothesis was the finding that when the same analysis was isolated to the units recorded from each monkey individually, repetition suppression was stronger in the monkey that was better at the task (monkey 2; *Figure 5c–d*).

## Predicting behavioral response patterns from neural signals

To quantitatively assess whether IT neural signals could account for the monkeys' behavioral reports, we applied two types of linear decoding schemes to the IT data. The first, a spike count classifier (SCC), is an instantiation of the strictest form of the repetition suppression hypothesis in that it differentiated novel versus familiar responses based on the total number of spikes across the IT population (i.e. every unit in the population received a weight of 1). The second, a Fisher Linear Discriminant (FLD), is an extension of the SCC that allows for IT units to be differentially weighted and allows for weights to be positive as well as negative (corresponding to repetition suppression and enhancement, respectively).

Because the neural data collected in any individual recording session had too few units to fully account for the monkeys' behavior (e.g. near 100% correct for 1-back familiar images), we concatenated units across sessions to create a larger pseudopopulation, where responses were quantified 150–400 ms following stimulus onset. When creating this pseudopopulation, we aligned data across sessions in a manner that preserved whether the trials were presented as novel or familiar as well as their n-back separation. More specifically, the responses for each unit always contained sets of novel/familiar pairings of the same images, and pseudopopulation responses across units were always aligned for novel/familiar pairs that contained the same n-back separation. Because different images were used in each session, aligning images this way implicitly assumes that the total numbers of spikes are matched across different images, the data recorded in any one session is a representative sample of those statistics, and that the responses of the units recorded in different sessions are uncorrelated. When the number of images in a session exceeded the number required to construct the pseudopopulation, a subset of images were selected randomly, and we confirmed that our main results did not change for different random selections. In the case of the pooled data, the resulting pseudopopulation consisted of the responses from 799 neurons to 107 images presented as both novel and familiar (i.e. 15, 15, 16, 17, 17, 15 and 12 trials at 1, 2, 4, 8, 16, 32 and 64-back, respectively).

We begin by illustrating our procedure for computing neural predictions of the behavioral forgetting functions and reaction time patterns with the FLD weighted linear read-out, applied to the data pooled across the two subjects. We then present a more systematic comparison between

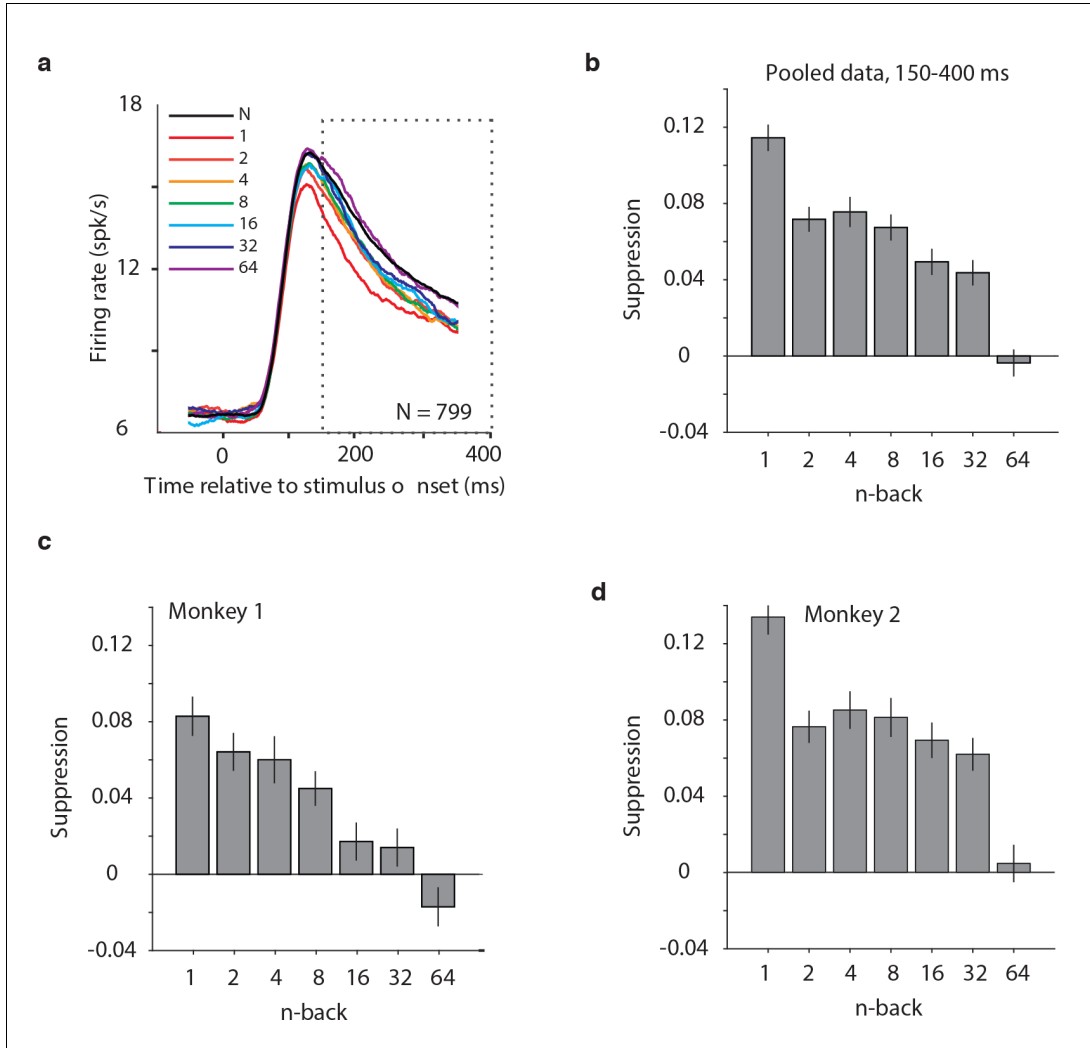

**Figure 5.** Average IT repetition suppression magnitudes. (**a**) Grand mean firing rates for all units, plotted as a function of time aligned to stimulus onset, parsed by images presented as novel (black) versus familiar at different n-back (rainbow). Traces were computed in 1 ms bins and smoothed by averaging across 50 ms. The dotted box indicates the spike count window corresponding to the analysis presented in panels (**b–d**). The absence of data at the edges of the plot (−50:−25 ms and 375:400 ms) reflects that the data are plotted relative to the centers of each 50 ms bin and data were not analyzed before −50 ms or after the onset of the go cue at 400 ms. (**b**) The calculation of suppression magnitude at each n-back began by quantifying the grand mean firing rate response to novel and familiar images within a window positioned 150 ms to 400 ms after stimulus onset. Suppression magnitude was calculated separately for each n-back as (novel – familiar)/novel. (**c–d**) Suppression magnitudes at each n-back, computed as described for panel b but isolated to the units recorded in each monkey individually. Error reflects SEM.

DOI: https://doi.org/10.7554/eLife.32259.008

different decoders applied to each monkey's individual data. To compute neural predictions for behavioral forgetting functions, we began by training an FLD linear decoder to discriminate the same images presented as novel versus as familiar (***Figure 6a***) using the data corresponding to all n-backs simultaneously. The FLD training procedure assigned a weight to each neuron proportional to the amount of linearly-separable visual memory information reflected in its responses (i.e. it's d'; see Materials and methods), and a single criterion value to parse the combined, weighted population responses for novel versus familiar predictions. A final parameter specified the size of the IT population under consideration (detailed below). Shown in ***Figure 6b*** are the neural estimates of the distributions of memory signal strength at each n-back, computed across many iterations of the

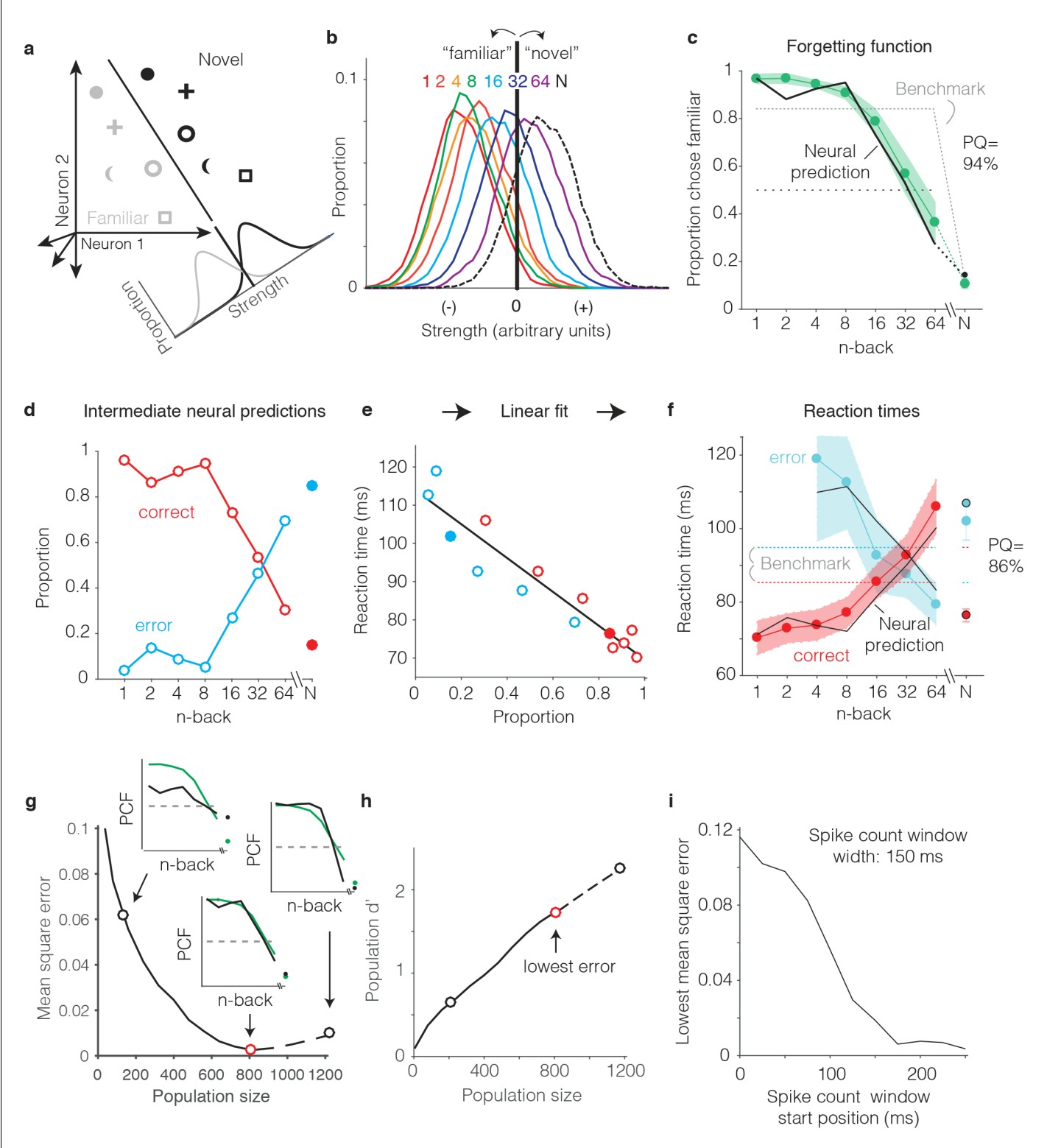

**Figure 6.** Transforming IT neural data into behavioral predictions. In all panels, behavioral and neural data correspond to the data pooled across the two monkeys and methods are illustrated through application of only one linear decoder (the FLD). (a) A cartoon depiction of how memory strength was measured for each n-back. Shown are the hypothetical population responses of 2 neurons to different images (represented by different shapes) shown as novel (black) versus as familiar (gray). The line depicts a linear decoder decision boundary optimized to classify images as novel versus as familiar. Distributions across images within each class are calculated by computing the linearly weighted sum of each neuron's responses and

*Figure 6 continued on next page*

*Figure 6 continued*

subtracting a criterion. (**b**) Distributions of the linearly weighted IT population response, as a measure of memory strength, shown for novel images (black dotted) and familiar images parsed by n-back (rainbow), for a population of 799 units. To compute these distributions, a linear decoder was trained to parse novel versus familiar across all n-back via an iterative resampling procedure (see Materials and methods). (**c**) Black: the neural prediction of the behavioral forgetting function, computed as the fraction of each distribution in panel b that fell on the 'familiar' (i.e. left) side of the criterion. Behavioral data are plotted with the same conventions as *Figure 3a,c*. Prediction quality (PQ) was measured relative to a step function benchmark (gray dotted) with matched average performance (see text). (**d**) The first step in the procedure for estimating reaction times, shown as a plot of the proportions of each distribution from panel b predicted to be correct versus wrong, as a function of n-back. Solid and open circles correspond to novel and familiar trials, respectively. Note that the red curve (correct trials) simply replots the predictions from panel c and the blue curve (error trials) simply depicts those same values, subtracted from 1. (**e**) A plot of the proportions plotted in panel d versus the monkeys' mean reaction times for each condition, and a line fit to that data. (**f**) The final neural predictions for reaction times, computed by passing the data in panel d through the linear fit depicted in panel e. Behavioral data are plotted with the same conventions as *Figure 3b,d*. Also shown are the benchmarks used to compute PQ (labeled), computed by passing the benchmark values showing in panel c through the same process. (**g**) Mean squared error between the neural predictions of the forgetting function and the actual behavioral data, plotted as a function of population size. Solid lines correspond to the analysis applied to recorded data; the dashed line corresponds to the analysis applied to simulated extensions of the actual data (see *Figure 6—figure supplement 1*). Insets indicate examples of the alignment of the forgetting function and FLD neural prediction at three different population sizes, where green corresponds to the actual behavioral forgetting function and black corresponds to the neural prediction. PCF = proportion chose familiar. The red dot indicates the population size with the lowest error (n = 799 units). (**h**) Overall population d' for the novel versus familiar task pooled across all n-back, plotted as a function of population size with the highlighted points from panel g indicated. (**i**) The analysis presented in panel g was repeated for spike count windows 150 ms wide shifted at different positions relative to stimulus onset. Shown is the minimal MSE for each window position. All other panels correspond to spikes counted 150–400 ms.

DOI: https://doi.org/10.7554/eLife.32259.009

The following figure supplement is available for figure 6:

**Figure supplement 1.** Extrapolating SCC and FLD predictions to larger sized populations.
DOI: https://doi.org/10.7554/eLife.32259.010

cross-validated linear classifier training and testing procedure for the best sized population (n = 799 units). As expected, we found that the weighted population response strengths were largest for novel images (*Figure 6b*, black) and were weakest for familiar images presented as immediate repeats (*Figure 6b*, red). Between these two extremes, we observed a continuum of strengths loosely organized according by n-back (*Figure 6b*, rainbow). Finally, a neural prediction for the forgetting function was computed as the fraction of each distribution that fell on the 'familiar' side of the criterion differentiating novel versus familiar predictions (*Figure 6c*). This analysis revealed a high degree of alignment between the neural prediction at each n-back and behavior, including high performance for familiar images presented at low n-back, performance at mid-range n-back that fell off with a similar sensitivity, and performance at the longest n-back (64) that fell below chance (*Figure 6c*). Similarly, neural predictions for novel images were well-aligned with the monkeys' behavioral reports (*Figure 6c*, 'N').

To produce neural predictions for reaction times, we turned to strength theory (*Figure 4*). Shown in *Figure 6d* is the first step required for making those predictions: a plot of the neural predictions for the proportions of 'correct' and 'error' trials, plotted as a function of n-back. Note that the correct predictions simply replicate the forgetting function shown in *Figure 6c*, and the error predictions are simply those same values, subtracted from 1. While these plots directly follow from *Figure 6c*, we include them to illustrate that they qualitatively reflected an inverted version of the monkeys' behavioral reaction time plots, including an x-shaped pattern. To determine the degree to which these qualitative relationships quantitatively predict the monkeys' reaction times, we examined the relationship between the proportions plotted in *Figure 6d* and the monkeys' mean reaction times, and found it to be approximately linear (*Figure 6e*). We thus fit a two parameter linear function to convert the neural predictions of these proportions into reaction times (*Figure 6e*, black line). The resulting neural predictions were largely aligned with the monkeys' mean reaction times (*Figure 6f*), including increasing reaction times as a function of n-back on correctly reported familiar trials, decreasing reaction times as a function of n-back on familiar trials in which the monkeys' made errors, and the characteristic x-shaped pattern. Additionally, shorter mean reaction times for novel images on correct versus error trials were largely well-predicted by the neural data.

One important step in the procedure, not detailed above, involved determining the appropriate IT population size for making neural and behavioral comparisons. Because there really wasn't a way

to do this *a priori*, we applied a fitting approach in which we computed the mean squared error (MSE) between the actual forgetting functions and their neural predictions at a range of population sizes, including simulated extensions of our population up to sizes 50% larger than the maximal size we recorded (*Figure 6—figure supplement 1*). The existence of a minimum in these plots follows from the fact that they depict the error between the neural prediction and the behavioral forgetting function (as opposed to overall neural population d' for this task, which continued to increase with increasing population size; *Figure 6h*). When too few units were included in the population, neural d' was too low and high performance at low n-back was underestimated (*Figure 6g*, left inset). In contrast, when too many units were included in the population, neural population d' was too high and performance at low n-back was over-saturated (*Figure 6g*, right inset). Additionally, for populations that were too large, performance fell off with n-back with a slope that was too steep. Of interest was the question of whether a global alignment of behavioral and neural sensitivity produced an accurate neural prediction of the shape for forgetting function with n-back. In the case of the FLD applied to the pooled data, the best population size fell near the maximal size of the total number of units that we recorded (n = 799, *Figure 6g*, red dot). The analyses presented thus far were computed based on spike count windows 150–400 ms following stimulus onset. A complementary plot illustrates how the position of the spike count window relative to stimulus onset impacted the best MSE (across all population sizes) for spike count windows 150 ms wide (*Figure 6i*). Consistent with the arrival of a visual memory signal that is delayed relative to onset but remains relatively constant thereafter, error was high for windows that began earlier than 150 ms following stimulus onset and then saturated. This suggests that the 150–400 ms position of the spike count window used to analyze the data throughout this report was a reasonable selection.

As a final step for our procedure, we determined a measure of prediction quality for both the forgetting function and reaction time patterns. Our measure benchmarked the MSE between the behavioral patterns and neural predictions by the worst-possible fit given that our procedure involves a global alignment of behavioral and neural data (*Figure 6g*). The upper bound of our measure, 100% 'prediction quality' (PQ), reflects a neural prediction that perfectly replicates the behavioral data. The lower bound (0% PQ) was computed as the MSE between the actual behavioral function and a predicted forgetting function that took the shape of a step, matched for global performance (percent correct across all conditions; *Figure 6c,f*, dotted). The rationale behind the step is that under a reasonable set of assumptions (i.e. that performance as a function of n-back should be continuous, have non-positive slope, and be centered around chance), a step reflects the worst possible fit of the data. Finally, PQ was calculated as the fractional distance of the MSE between these two benchmarks. In the case of the FLD applied to the pooled data, PQ was 94% for the forgetting function and 86% for the reaction time data (*Figure 6c,f*). We emphasize that these numbers reflect the quality of generalized neural predictions to the behavioral reports, as these neural predictions were not fit directly to the behavioral data in a manner not already accounted for by the PQ measure.

## A weighted linear read-out of IT more accurately predicted behavior than a total spike count decoding scheme

Our methods for determining predictions of the SCC decoder differed only in the algorithm used to combine the spike counts across the population into a measure of memory strength (*Figure 6b*). In the case of the SCC, the weight applied to each unit was 1, and the training procedure determined a single criterion value to parse the total population spike counts into novel versus familiar predictions. The same cross-validated procedure used for the FLD was applied to the SCC to determine distributions analogous to those depicted in *Figure 6b*. When applied to the data pooled across the two monkeys, the best sized SCC decoded population was 559 units (*Figure 7a*). Additionally, we found that while the SCC was a better predictor of behavior than the FLD for smaller sized populations (less than 400 neurons), the FLD was a better predictor of behavior overall (*Figure 7a*). Examination of a plot of overall population d' as a function of population size (*Figure 7b*) reveals that the minimal error fell at the same population d' for both decoding schemes, consistent with the notion that our procedure involved a global matching of overall performance between the behavioral and neural data. The fact that the lowest MSE differed between the two decoding schemes reflects differences in the shapes of the neural predictions following global performance matching. *Figure 7b* also reveals systematically better global performance of the SCC as compared to the FLD for

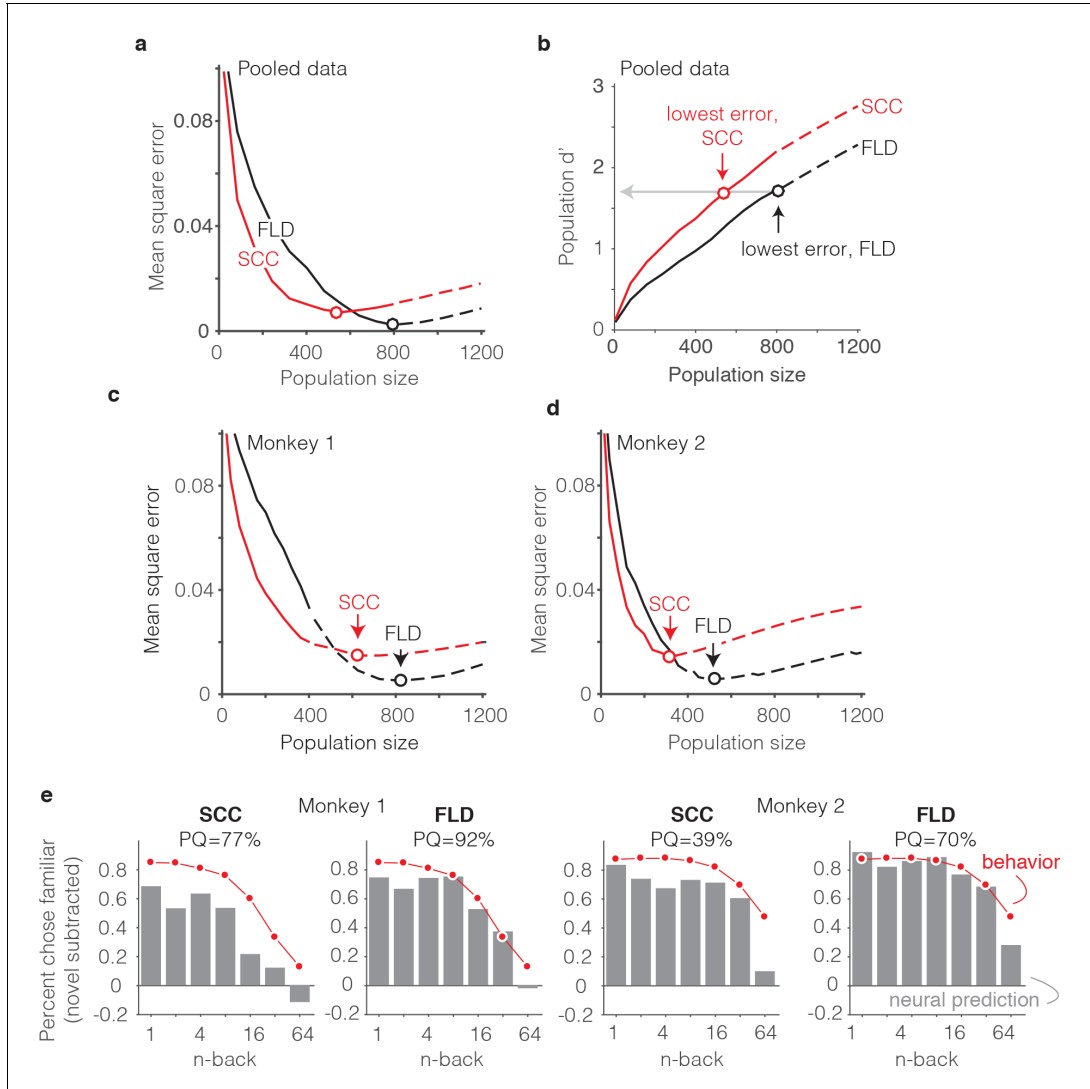

**Figure 7.** The FLD decoder is a better predictor of behavioral performance than the SCC. (**a**) Plot of mean square error as a function of population size, computed as described for *Figure 6g* for the data pooled across both monkeys, and shown for both the FLD (black) and SCC (red) decoders. Dots correspond to the population size with the smallest error (FLD = 799 units; SCC = 625 units). (**b**) Plot of overall population d' computed as described in *Figure 6h* but shown for both the FLD and SCC decoders. Dots correspond to the same (optimal) population sizes indicated in panel a. (**c–d**) The same analysis shown in panel a, but isolated to the data collected from each monkey individually. Best population sizes, Monkey 1: FLD = 800 units; SCC = 625 units; Monkey 2: FLD = 525 units; SCC = 316 units. (**e**) Gray: predicted forgetting functions, computed as described for *Figure 6c*, but plotted after subtracting the false alarm rate for novel images (i.e. a single value across all n-back). Red: the actual forgetting functions, also plotted after subtracting the novel image false alarm rate. These plots are a revisualization of the same data plotted before false alarm rate subtraction in *Figure 7—figure supplement 1a-b*. PQ: prediction quality, computed as described in panel 6 c.

DOI: https://doi.org/10.7554/eLife.32259.011

The following figure supplement is available for figure 7:

**Figure supplement 1.** FLD and SCC predictions for each monkey.

DOI: https://doi.org/10.7554/eLife.32259.012

matched sized populations, which is likely a consequence of the fact that a smaller number of parameters are fit with the SCC read-out and the estimation of FLD weights is a noisy process.

A comparison of SCC and FLD MSE plots isolated to each monkey's data revealed that the FLD decoder was a better predictor of behavior in both individuals (*Figure 7c–d*). Why was the FLD weighted linear decoder a better predictor of the behavioral forgetting function? This was because the spike count decoding scheme under-predicted memory strength, particularly at the longest

delays. While this is discernable in plots of the raw alignment of the behavioral and neural data for each monkey plotted with the same conventions as *Figure 6f* (*Figure 7—figure supplement 1a–b*), it is more easily observed in a visualization of the data in which the proportion of familiar choices for both the behavioral data and neural predictions are plotted after subtracting the false alarm rate for the novel images (*Figure 7e*), thus producing plots analogous to the suppression plots presented in *Figure 5c–d*. For example, in monkey 1, the SCC decoder predicted that the monkey would report 64-back familiar images as familiar at a rate lower than the false alarm rate for novel images, whereas the actual forgetting function reflected a small amount of remembering after a 64-back delay (*Figure 7e*). Similarly, in monkey 2, the SCC predicted rate of remembering at 64-back under-predicted the actual rate reflected in the behavior (*Figure 7e*). In contrast, the FLD better predicted the behavior across all n-back in both animals (*Figure 7e*). Lower MSE for the FLD as compared to SCC translated into higher neural PQ in each monkey (*Figure 7e* – labeled; not shown for the pooled data: SCC PQ = 83%, FLD PQ = 94%). The same behavioral and neural comparisons, plotted with the same conventions as *Figure 6c* and *Figure 6f*, are shown in *Figure 7—figure supplement 1*. We note that while the FLD PQ was lower in monkey two as compared to monkey 1 (monkey 1 FLD PQ = 92%, monkey 2 FLD PQ = 70%), this was not due to a lower MSE of the fits in monkey 2 (*Figure 7c–d*) but rather due to the fact that the forgetting function for monkey two better resembled the step benchmark for computing PQ, thus reducing the PQ dynamic range (*Figure 7—figure supplement 1a–b*).

Together, these results suggest that a weighted linear read-out was a better description of the transformation between IT neural signals and single-exposure visual memory behavior than a total spike count decoding scheme.

## The single-unit correlates of the weighted linear decoding scheme

The results presented above suggest that the SCC under-predicted memory strength as a function of n-back whereas the FLD prediction was more accurate. At least two different scenarios might lead to this result. First, it could be the case that visual memory may be reflected as net repetition suppression in some units and net repetition enhancement in others (across all n-back). In this scenario, the FLD would preserve both types of memory information (by assigning positive and negative weights for enhancement and suppression, respectively), whereas these two types of effects would cancel in a SCC decoding scheme, resulting in information loss. Alternatively, it might be the case that the repetition suppression hypothesis is approximately correct insofar as the IT units that carry visual memory signals systematically reflect visual memory with net repetition suppression, however, repetition suppression may be stronger at longer n-back for some units than others. In this scenario, better FLD behavioral predictions would result from preferentially weighting the neurons with the strongest (by way of longest lasting) visual memory signals. As described below, our results suggest that the latter scenario is a better description of our data.

To distinguish between these two scenarios, we began by examining the distributions of unit d' as a proxy for the FLD decoding weights. In both monkeys, the unit d' means were significantly shifted toward positive values (*Figure 8a–b*; Wilcoxon sign rank test, monkey one mean = 0.05, $p=6.8*10^{-17}$; monkey two mean = 0.12, $p=1.9*10^{-41}$). In both monkeys, units with negative d' were also present (proportion of negative units for monkey 1 = 32%; monkey 2 = 19%), although from raw d' values alone, the degree to which negative d' resulted from reliable net repetition enhancement versus from noise is unclear. A comparison of the mean responses to novel as compared to familiar images for each unit revealed that very few units with negative d' had statistically distinguishable responses (bootstrap statistical test; criterion $p<0.01$; monkey 1: positive d' units = 14; negative d' units = 3; monkey 2: positive d' units = 75; negative d' units = 2). While a screen of $p<0.01$ can under-estimate the contributions of a unit to population performance, additional analyses, described below, confirm that negative d' units made a measurable but modest contribution to the differences between the SCC and FLD behavioral predictions.

To understand how these unit d' measures combined to determine behavioral predictions, we performed an analysis to determine the minimal number of 'best' d' IT units required to predict behavior. The general idea behind this analysis is that if it were the case that strong signals were carried by a small subpopulation of units, error should plateau quickly when only best units are included. We thus compared FLD behavioral prediction error trajectories for the pooled data (to maximize the numbers of directly measured units) when subsets of units were randomly sampled

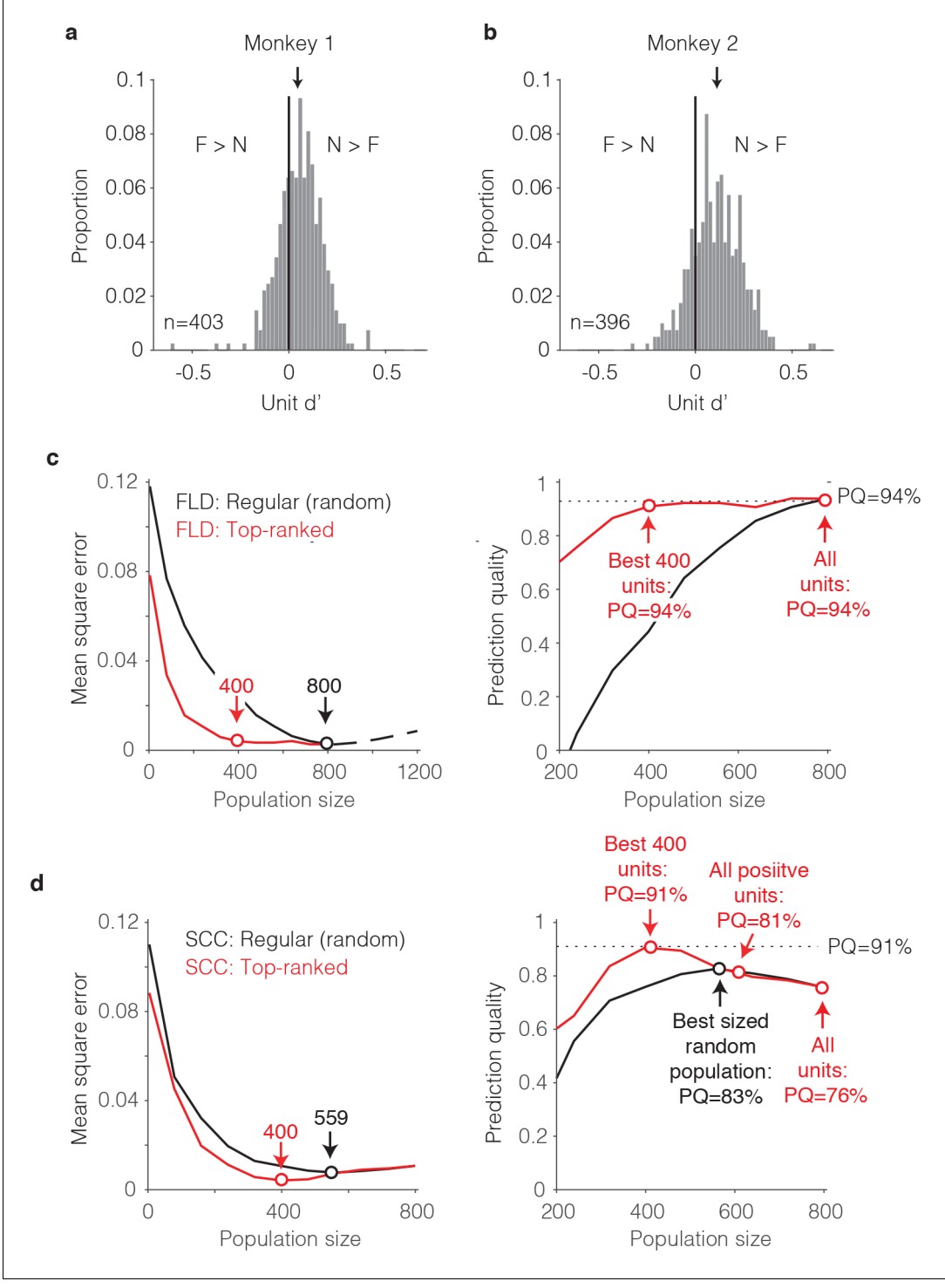

**Figure 8.** The single-unit correlates of the weighted linear decoding scheme. (**a–b**) Distributions of unit d',
computed for each monkey. Arrows indicate means. (**c**) *Left,* Comparison of behavioral prediction error trajectories
for an FLD decoder applied to randomly selected units (replotted from *Figures 6g* and *7a*) versus when the top-
ranked units for each population size were selected before cross-validated testing. Dots correspond to population
sizes with lowest error. *Right,* Conversion of behavioral error predictions (MSE) into prediction quality (PQ). Dots
indicate PQ for the best sized population and for all units. (**d**) *Left,* Comparison of behavioral prediction error
trajectories for an SCC decoder applied to randomly selected units (replotted from *Figure 7a*) versus when the
top-ranked units for each population size were selected before cross-validated testing. Dots correspond to
population sizes with lowest error. *Right,* Conversion of MSE into PQ. Black dot indicates PQ for the best sized
*Figure 8 continued on next page*

*Figure 8 continued*

population for randomly selected units. Red dots indicate PQ for the 400 top-ranked, positive sign (d'>0) units, all positive sign units, and all units.

DOI: https://doi.org/10.7554/eLife.32259.013

(our typical procedure) versus when the top-ranked d' units were selected via a cross-validated procedure (i.e. based on the training data; *Figure 8c*, left). We also converted these MSE measures into prediction quality estimates (*Figure 8d*, right). We found that 400 top-ranked IT units were required to achieve the same prediction quality as 800 randomly sampled units, suggesting that FLD behavioral predictions rely on visual memory signals that are distributed across approximately half of the IT population. The absence of a contribution from the lower-ranked 50% of the IT population could not be attributed to non-responsiveness, as nearly all the units (759/799, 95%) produced statistically significant stimulus-evoked responses that differed from the pre-stimulus baseline period (bootstrap statistical test; criterion $p<0.01$; comparison of spike count windows ($-150$–0) ms versus (75 – 225) ms relative to stimulus onset).

Why did the FLD produce better behavioral predictions than the SCC (*Figure 7a*)? To address this question, we repeated the top-ranked analysis for the SCC. Specifically, we performed a cross-validated procedure in which units were ranked by their signed d' as described above for the ranked FLD, but within the top-ranked units, spikes were summed to produce behavioral predictions (*Figure 8d*). One can envision this as a binary classifier where the top-ranked units each receive a weight of 1 whereas the remaining units each receive a weight of 0. Surprisingly, the ranked-SCC decoder also peaked at 400 units and performed nearly as well as the ranked-FLD (ranked SCC PQ for 400 units = 91%, *Figure 8d*; ranked FLD PQ for 400 units = 94%, *Figure 8d*). This suggests that within the subset of 50% top-ranked IT units, spikes could largely be summed to make behavioral predictions.

What happens when the 50% bottom-ranked units are added to each type of decoder? Addition of bottom-ranked units had no impact on the ranked-FLD (*Figure 8c* right, 'All units'). This suggests that the FLD largely disregards the lower 50% ranked units when making behavioral predictions. In contrast, the introduction of the lower 50% ranked units detrimentally impacted ranked-SCC behavioral predictions (ranked SCC PQ for best 50% of units = 91%; for all units = 76%; *Figure 8d*, right). This is presumably because the SCC does not have a weighting scheme and was thus forced to incorporate them. When parsed by the sign of d' for the lower-ranked units, addition of lower-ranked, positive d' units reduced ranked-SCC behavioral predictions from 91% to 81%, and further addition of negative d' units reduced behavioral predictions to 76% (*Figure 8d*, right). Returning to the two scenarios presented at the beginning of this section, these results suggest that better FLD as compared to SCC behavioral predictions could largely be attributed to the FLD preferentially weighting the neurons with the strongest (by way of longest lasting) visual memory signals, as opposed to the inability of the SCC to appropriately weight reliable, mixed sign modulation (i.e. mixtures of repetition suppression and enhancement).

Together, these results suggest that largely accurate behavioral predictions could be attributed to ~50% of IT units whose memory signals were reflected as repetition suppression, and within this top-ranked subpopulation, spike counts could largely be summed. These results also show that while the lower ranked units had a detrimental impact on the ability of the spike count decoder to produce accurate behavioral predictions, a weighted linear decoder largely disregarded these otherwise confounding responses.

## The impact of visual selectivity on population size

As a complementary consideration, we also examined the impact of visual selectivity on the size of the population required to account for behavior. Hypothetically, if only a small fraction of IT units were activated in response to any one image, a large population would be required to support robust visual memory behavioral performance. Because our data only include the response to each image twice (once as novel and repeated as familiar), and measures of visual selectivity (e.g. 'sparseness') produce strongly biased estimates with limited samples (*Rust and DiCarlo, 2012*), we applied

a simulation-based approach to determine how visual selectivity impacted the population size required to make accurate behavioral predictions.

The general idea behind this analysis is to compare the best population size for our intact data with a simulated version of our data in which visual memory signals have been kept intact but visual selectivity has been removed. To perform this analysis, we began by creating a simulated 'replication' population designed to match the image selectivity, memory signal strength, and grand mean spike count response for each unit we recorded, followed by the introduction of Poisson trial variability (see Materials and methods). This simulated population produced FLD behavioral prediction error trajectories that were highly similar to the intact population, both when computed with the regular FLD (*Figure 9a*, gray versus black), as well as with the ranked-FLD (*Figure 9b*, gray versus black), suggesting that the simulation was effective at capturing relevant aspects of the raw data. Next, we created a simulated 'visual-modulation-removed' version of each unit in which the memory signal strength (as a function of n-back) and the grand mean spike count response (across all conditions) were preserved, but visual selectivity was removed (see Materials and methods). Conceptually, one can think about this simulation as creating a version of each unit with pure selectivity for visual memory in the absence of visual modulation. The FLD behavioral prediction error trajectory of the visual-modulation-removed population fell faster than the replication population and took on approximately the same MSE as the intact population with only 479 (as compared to 800) units for the regular FLD (*Figure 9a*, red) and only 159 (as compared to 400) units for the ranked-FLD (*Figure 9b*, red). These results suggest that visual selectivity resulted in a substantial increase in the number of units required to account for behavioral performance within the FLD decoding scheme.

In sum, at least two factors combined to determine that a large number of FLD decoded IT units (~800) were required to accurately predict single-exposure behavioral performance. First, the visual memory signals that combined to produce largely accurate behavioral predictions were limited to ~50% of the IT population. Second, as a consequence of visual selectivity, the presentation of an image only activated a subset of units, thus increasing the population size required for robust neural performance that was capable of generalizing to new images.

## Individual behavioral patterns were reflected in the IT neural data

As a final, complementary set of analyses, we focused on the neural correlates of the differences in behavioral patterns reflected between the two animals. From the results presented above, we can infer that this is not a straightforward relationship: while the animal that was better at the task

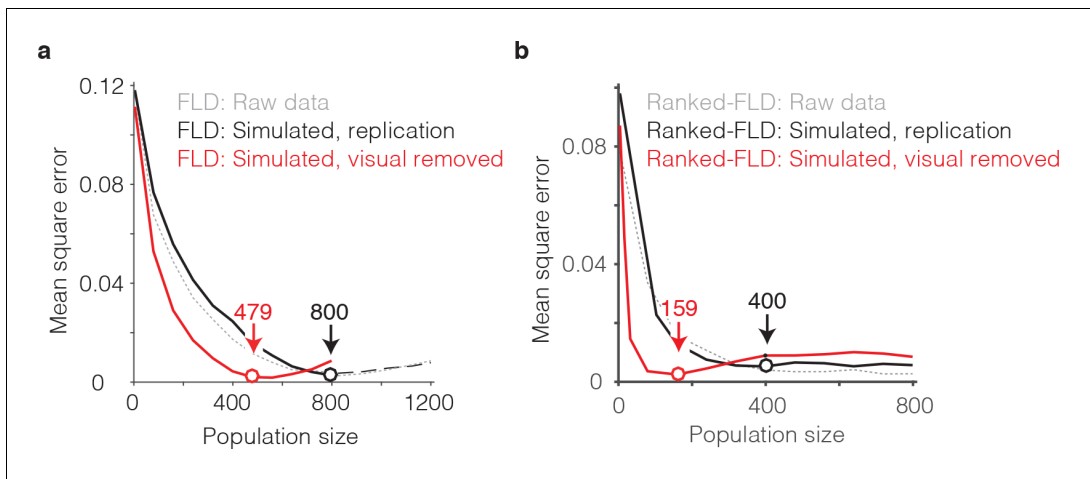

**Figure 9.** The impact of visual selectivity. (**a**) FLD behavioral prediction error trajectories for the actual data (gray, replotted from *Figure 6g*), a simulated replication of the data in which both the visual selectivity and the visual memory signals for each unit were replicated (black), and a simulated version of the data in which the visual memory signals were preserved for each unit but visual selectivity was removed (see Materials and methods). (**b**) The same three FLD behavioral prediction error trajectories, computed with a ranked-FLD.
DOI: https://doi.org/10.7554/eLife.32259.014

(monkey 2, *Figure 3a,c*) had stronger average repetition suppression (*Figure 5c–d*), fewer units were also required to account that animal's behavior (500 versus 800, *Figure 7c–d*). This suggests that differences in behavioral performance between the two monkeys does not simply reflect two populations that are matched in size but contain neurons whose visual memory signals differ in average strength. For deeper insights into the differences between animals, we performed an analysis in which we attempted to predict each monkey's behavioral forgetting functions from the other monkey's neural data using the FLD decoder (*Figure 10a–b*). For both monkeys, the minimal error (as a function of population size) was lower when behavioral and neural data came from the same monkey as compared to when they were mixed between monkeys (*Figure 10a–b*, red versus black dots) and this translated to better PQ when behavioral and neural data came from the same animal versus when they came from different animals (*Figure 10c*).

*Figure 10c* illustrates the alignment of the behavioral forgetting functions and their neural predictions, after subtracting the false alarm rate for novel images (similar to 7e), shown for the cases in which behavioral and neural data came from the same animal and when they were crossed. In the case of monkey 1, the neural prediction from the same animal largely captured the pattern of forgetting with n-back, whereas the neural data from monkey two predict a shape that was too flat. In other words, FLD applied to the neural data from monkey two predicted a similar amount of forgetting across a wide range of n-back and this pattern was inconsistent with the steeper fall-off in that same range reflected in the behavior of monkey 1 (*Figure 10c*, monkey 1 'Cross'). Similarly, the neural data collected from monkey one reflected a considerable amount of forgetting at higher n-back, whereas the behavioral data from monkey two were more flat in this range. This led to a discrepancy between the behavioral data and neural predictions when aligned around the novel image prediction (*Figure 10c*, monkey 2 'Cross').

While our study was limited to only two subjects and thus lacked the power to establish individual differences, the better alignment of behavioral and neural data within subjects versus across subjects

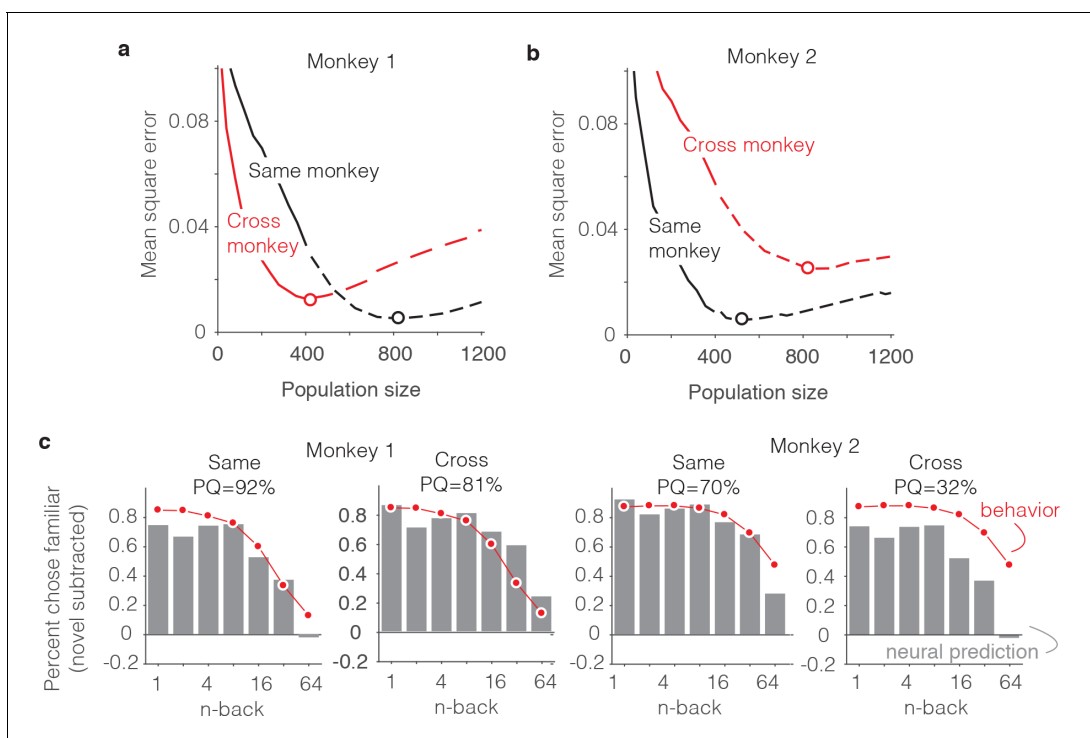

**Figure 10.** Alignment of individual behavioral and neural data. (a–b) Plot of mean squared error as a function of population size, computed as described for *Figure 6g* but compared when behavioral and neural data come from the same monkey (black) versus when behavioral and neural data are crossed between monkeys (red). (c) Comparison of predicted and forgetting functions, plotted after subtracting the false alarm rate for novel images as in *Figure 7e*, for population sizes indicated by the dots in panels a-b. PQ = prediction quality.
DOI: https://doi.org/10.7554/eLife.32259.015

is an effective demonstration that signal strength and population size cannot simply be traded off to fit any possible behavioral function. Additionally, these results provide added support of the hypothesis that single-exposure visual memory behaviors are in fact reflected in the neural responses of IT cortex.

## Discussion

This study was designed to test the hypothesis that the signals supporting single-exposure visual recognition memories, or equivalently answers to the question, 'Have I seen that image before?", are reflected as decrements in the responses of neurons in IT with stimulus repetition (*Fahy et al., 1993*; *Li et al., 1993*; *Miller and Desimone, 1994*; *Riches et al., 1991*; *Xiang and Brown, 1998*). Prior to this study, this hypothesis had received mixed support from human fMRI studies (*Gonsalves et al., 2005*; *Turk-Browne et al., 2006*; *Ward et al., 2013*; *Xue et al., 2011*) and was largely untested at the resolution of individual neurons. We found that a strict interpretation of the repetition suppression hypothesis in the form of counting the total numbers of spikes across the IT population provided an incomplete account of single-exposure visual memory behavior (*Figure 7*), whereas a weighted linear read-out of IT provided reasonably accurate predictions of the rates of forgetting as a function of time (*Figure 6c*, *Figure 7e*), as well as mean reaction time patterns (*Figure 6f*; *Figure 7—figure supplement 1*). Additionally, behavioral predictions could be attributed to IT visual memory signals that were reflected as repetition suppression (*Figure 8*) and were intermingled with visual selectivity (*Figure 9*), but only when combined across the most sensitive 50% of IT units (*Figure 8c–d*).

Our study was focused on changes in IT that follow a single image exposure, and the net repetition suppression that we observed is qualitatively consistent with earlier reports (*Fahy et al., 1993*; *Li et al., 1993*; *Miller and Desimone, 1994*; *Riches et al., 1991*; *Xiang and Brown, 1998*). Net repetition suppression has also been reported following exposure of IT neurons to the same images hundreds or thousands of times (*Anderson et al., 2008*; *Baker et al., 2002*; *Freedman et al., 2006*; *Lim et al., 2015*; *Meyer et al., 2014*; *Woloszyn and Sheinberg, 2012*). However, the suppression that we observed was transient (~5 min), whereas the suppression that follows many repeated image exposures is much longer lasting. Some studies have reported repetition enhancement in IT for images that are highly familiar, particularly when an image falls at the peak of a neuron's tuning function and the neuron in question is excitatory (*Lim et al., 2015*; *Woloszyn and Sheinberg, 2012*). In our study, we found no evidence that net repetition enhancement contributed to behavioral predictions. At the next stage of processing in the medial temporal lobe, perirhinal cortex, there are indications that following many repeated exposures, the sign of familiarity modulation may flip from net suppression to net enhancement (*Landi and Freiwald, 2017*; *Tamura et al., 2017*). In contrast, following a limited number of exposures, neurons in a region now attributed to perirhinal cortex have been reported to exhibit repetition suppression (*Li et al., 1993*; *Miller et al., 1991*). Future work will be required to determine the effects of image familiarity in IT and perirhinal cortex as images transition from novel to highly familiar.

Notably, when monkeys are engaged in a task that involves both stimulus repetition as well as a same/different judgment about repeated stimuli, heterogeneous combinations of repetition enhancement and suppression are observed in IT and perirhinal cortex (*Miller and Desimone, 1994*; *Pagan et al., 2013*; *Vogels and Orban, 1994*). These results may reflect the fact that the responses of neurons in these brain areas reflect mixtures of the signals supporting visual memory, attention, and decision processes. In fact, considerable evidence supports the notion that the task a subject is engaged in at the time of viewing will have an impact on what will be remembered (reviewed by *Chun and Turk-Browne, 2007*). In our study, the targets were present at stimulus onset for the first monkey but delayed until the go cue (400 ms) in the second animal, and poorer performance of monkey one in this task may reflect divided attention between the visual image and the targets.

The neural correlates of explicit visual memory reports have been investigated in the human brain using PET (*Vandenberghe et al., 1995*) and fMRI (*Gonsalves et al., 2005*; *Turk-Browne et al., 2006*; *Ward et al., 2013*; *Xue et al., 2010*). A number of factors might contribute to the discrepancy between our study and human fMRI studies that fail to find a relationship between repetition suppression magnitudes in high-level visual brain areas and explicit visual memory reports (*Ward et al., 2013*; *Xue et al., 2011*). For example, one implication of our results is that near-single

unit resolution is required to determine how to appropriately weight IT units to account for single-exposure visual memory behaviors. In contrast, measures that average the responses across large numbers of neurons result in an information loss that cannot fully be recovered (e.g. via a multi-voxel pattern analysis). Another factor that may contribute to differences between our results and those studies is a distinct difference in experimental design: our study correlates repetition suppression and behavioral reports on the same trial, whereas these studies correlate repetition suppression to a second viewing of an image with the behavioral report about remembering during a third viewing. The rationale behind the fMRI design is a desire to dissociate memory processes from the processes involved in decision making and response execution. In our study, we were focused on evaluating the plausibility that the signal supporting visual memory behavioral reports is reflected in IT cortex, as opposed to the plausibility that memory signals are reflected in IT in the absence of a subject being engaged in a memory task. The consistent (positive) sign of the linear weights recovered across IT units suggests that our results cannot be accounted for by motor responses, as the task required the monkeys to saccade to two different targets to report novel versus familiar predictions and a motor account would require that all the IT neurons were tuned for the same target (e.g. 'upward' for monkey one and 'downward' for monkey 2). Finally, differences between our study and those reports could also arise from differences between species, analogous to the differences reported between monkey IT and human LOC for changes in the representations of highly familiar images as measured with fMRI (*Op de Beeck et al., 2006*; *Op de Beeck et al., 2008*).

Our results suggest that visual memory signals are reflected as repetition suppression in the majority of IT units and that reports of whether an image has been seen before can be predicted by counting the numbers of spikes across the top half of the repetition suppressed IT subpopulation (*Figure 8e*). One question not addressed in our experiments is how this type of decoding scheme could tease apart changes in total numbers of spikes due to stimulus repetition from changes in spike numbers due to other variables, such as contrast, luminance, object size, and potentially object identity (*Chang and Tsao, 2017*). In principle, the brain could address this by relying on neurons that are sensitive to visual memory but insensitive to these other types of variables. Future work will be required to investigate these issues.

Analysis of our reaction time patterns parsed by trial outcome (correct/error) revealed a characteristic x-shaped pattern (*Figure 3*) at odds with the predictions of standard models of decision making such as standard instantiations of the drift diffusion model. Extensions of the drift diffusion framework have been proposed in which reaction time asymmetries on correct versus error trials can be accounted for by adding per-trial noise in the decision variable drift rate or the decision variable start position (*Ratcliff and McKoon, 2008*). Our task was not designed to differentiate between these and other similar models, but rather to test the hypothesis that signals reflecting single-exposure visual memories are found in IT cortex. As such, we opted for the much simpler, lower-parameter description suggested by strength theory (*Murdock, 1985*; *Norman and Wickelgren, 1969*). The inverted relationship between proportion correct and reaction time captured by strength theory can loosely be thought of as a signature of confidence (e.g. when performance is higher, reaction times are faster), however, the drawback of strength theory is that it lends little biophysical insight into how this process might happen in the brain. Our study provides important constraints on models of the decision making process for single-exposure memory tasks, and should constrain future work in which this process is investigated more completely.

In this study, we adjusted the task parameters such that images were forgotten over minutes within sessions that lasted approximately one hour. This included reducing the viewing time from the longer durations used in previous human behavioral experiments (2–3 s) to ~400 ms. Our results suggest that forgetting rates are well-aligned between behavioral reports and IT neural signals within this regime. Will longer timescale memories be reflected by signals in IT as well? That remains to be seen. It could be the case that IT reflects single-exposure visual memories across all behaviorally-relevant timescales, alternatively, it could be the case that the signals reflecting single-exposure memories across longer timescales (e.g. hours and days) are only reflected in higher brain areas such as perirhinal cortex and/or the hippocampus.

A related issue is the question of where and how single-exposure visual memories are stored in the brain. Crucially, it is important to recognize that it does not necessary follow from the fact that a particular brain area reflects a memory signal, that it must be the locus at which storage occurs. It is likely the case that the visual memory signals that we observe are at least partially the consequence

of the cumulative adaptation-like processes that happen within IT and within brain areas preceding IT. What is less clear is whether these signals also reflect contributions from higher brain areas as well. Similarly, a computational description of the learning rule(s) that accurately capture the changes in the brain that follow a single image exposure remain to be determined. While important first steps toward those computational descriptions have been proposed (*Androulidakis et al., 2008*; *Lulham et al., 2011*) they have yet to be tested in deep neural network architectures that approximate the patterns of neural activity reflected in the visual system (e.g. *Yamins et al., 2014*).

## Materials and methods

Experiments were performed on two adult male rhesus macaque monkeys (*Macaca mulatta)* with implanted head posts and recording chambers. All procedures were performed in accordance with the guidelines of the University of Pennsylvania Institutional Animal Care and Use Committee under protocol 804222.

### The single-exposure visual memory task

All behavioral training and testing was performed using standard operant conditioning (juice reward), head stabilization, and high-accuracy, infrared video eye tracking. Stimuli were presented on an LCD monitor with an 85 Hz refresh rate using customized software (http://mworks-project. org).

As an overview of the monkeys' task, each trial involved viewing one image for at least 400 ms and indicating whether it was novel, (never seen before) or familiar (seen exactly once prior) with an eye movement to one of two response targets. Images were never presented more than twice (once as novel and then as familiar) during the entire training and testing period of the experiment. Trials were initiated by the monkey fixating on a red square (0.25°) on the center of a gray screen, within a square window of ±1.5°, followed by a 200 ms delay before a 4° stimulus appeared. The monkeys had to maintain fixation of the stimulus for 400 ms, at which time the red square turned green (go cue) and the monkey made a saccade to the target indicating that the stimulus was novel or familiar. In monkey 1, response targets appeared at stimulus onset; in monkey 2, response targets appeared at the time of the go cue. In both cases, targets were positioned 8° above or below the stimulus. The association between the target (up vs. down) and the report (novel vs. familiar) was swapped between the two animals. The image remained on the screen until a fixation break was detected.

The images used in these experiments were collected via an automated procedure that explored and downloaded images from the internet, and then scrubbed their metadata. Images smaller than 96*96 pixels were not considered. Eligible images were cropped to be square and resized to 256*256 pixels. An algorithm removed duplicate images. The resulting database included 89,787 images. Within the training and testing history for each monkey, images were not repeated. A representative sample of a subset of 49 images are presented in *Figure 2—figure supplement 1*.

The specific random sequence of images presented during each session was generated offline before the start of the session. The primary goal in generating the sequence was to select trial locations for novel images and their repeats with a uniform distribution of n-back (where n-back = 1, 2, 4, 8, 16, 32 and 64). This was achieved by constructing a sequence slightly longer than what was anticipated to be needed for the day, and by iteratively populating the sequence with novel images and their repeats at positions selected from all the possibilities that remained unfilled. Because the longest n-back locations (64) were the most difficult to fill, a fixed number of those were inserted first, followed by systematically working through the insertion of the same fixed number at each consecutively shorter n-back (32, 16 . . .). In the relatively rare cases that the algorithm could not produce that fixed number at each n-back, it was restarted. The result was a partially populated sequence in which 83% of the trials were occupied. Next, the remaining 17% of trials were examined to determine whether they could be filled with novel/familiar pairs from the list of n-back options (64, 32, 16-back . . .). For the very small number of trials that remained after all possibilities had been extinguished (e.g. a 3-back scenario), these were filled with 'off n-back' novel/familiar image pairs and these trials were disregarded from later analyses.

'Forgetting functions' (*Figure 3a,c* and *Figure 6c*) were computed as the mean proportion of trials each monkey selected the familiar target, across all trials and all sessions. Because behavioral outcome is a binary variable, error was estimated by computing the mean performance trace for each

session, and then computing the 97.5% confidence interval as 2.2*standard error of those traces. Mean reaction times (*Figure 3b,d* and *Figure 6f*) were computed as means across all trials and sessions, and 97.5% confidence intervals were computed as 2.2*standard error of those same values.

## Neural recording

The activity of neurons in IT was recorded via a single recording chamber in each monkey. Chamber placement was guided by anatomical magnetic resonance images in both monkeys. The region of IT recorded was located on the ventral surface of the brain, over an area that spanned 5 mm lateral to the anterior middle temporal sulcus and 14–17 mm anterior to the ear canals. Recording sessions began after the monkeys were fully trained on the task and after the depth and extent of IT was mapped within the recording chamber. Combined recording and behavioral training sessions happened 4–5 times per week across a span of 5 weeks (monkey 1) and 4 weeks (monkey 2). Neural activity was recorded with 24-channel U-probes (Plexon, Inc, Dallas, TX) with linearly arranged recording sites spaced with 100 μm intervals. Continuous, wideband neural signals were amplified, digitized at 40 kHz and stored using the Grapevine Data Acquisition System (Ripple, Inc., Salt Lake City, UT). Spike sorting was done manually offline (Plexon Offline Sorter). At least one candidate unit was identified on each recording channel, and 2–3 units were occasionally identified on the same channel. Spike sorting was performed blind to any experimental conditions to avoid bias. A multi-channel recording session was included in the analysis if: (1) the recording session was stable, quantified as the grand mean firing rate across channels changing less than 2-fold across the session; (2) over 50% of neurons were visually responsive (a loose criterion based on our previous experience in IT), assessed by a visual inspection of rasters; and (3) the number of successfully completed novel/familiar pairs of trials exceeded 100. In monkey 1, 21 sessions were recorded and 6 were removed (2 from each of the 3 criterion). In monkey 2, 16 sessions were recorded and 4 were removed (1, 2 and 1 due to criterion 1, 2 and 3, respectively). The sample size (number of successful sessions recorded) was chosen to approximately match our previous work (*Pagan et al., 2013*).

## Neural predictions of behavioral performance

Because the data recorded in any individual session (on 24 channels) corresponded to a population too small to provide a full account of behavioral performance, we combined data across sessions into a larger pseudopopulation (see Results). We compared the ability of four different linear decoders to predict the monkeys' behavioral responses from the IT pseudopopulation data. Spikes were counted in a window 150–400 ms following stimulus onset with the exception of *Figure 6i*, where spikes were counted in a 150 ms bin at sliding positions relative to stimulus onset.

For all decoders, the population response $x$ was quantified as the vector of simultaneously recorded spike counts on a given trial. To ensure that the decoder did not erroneously rely on visual selectivity, the decoder was trained on pairs of novel/familiar trials in which monkeys viewed the same image (regardless of behavioral outcome and for all n-back simultaneously). Here we begin by describing each decoder, followed by a description of the cross-validated training and testing procedure that was applied in the same manner to each one.

All four decoders took the general form of a linear decoding axis:

$$f(\mathbf{x}) = \mathbf{w}^T\mathbf{x} + b \tag{1}$$

where $\mathbf{w}$ is an N-dimensional vector (and N is the number of units) containing the linear weights applied to each unit, and b is a scalar value. What differed between the decoders was how these parameters were fit.

### Fisher Linear Discriminant variants (FLD, ranked FLD)

In the case of the FLD, the vector of linear weights was calculated as:

$$\mathbf{w} = \Sigma^{-1}(\mu_1 - \mu_2) \tag{2}$$

and b was calculated as:

$$b = \mathbf{w} \cdot \frac{1}{2}(\mu_1 + \mu_2) = \frac{1}{2}\mu_1^T\Sigma^{-1}\mu_1 - \frac{1}{2}\mu_2^T\Sigma^{-1}\mu_2 \tag{3}$$

Here $\mu_1 and \mu_2$ are the means of the two classes (novel and familiar, respectively) and the mean covariance matrix is calculated as:

$$\Sigma = \frac{\Sigma_1 + \Sigma_2}{2} \qquad (4)$$

where $\Sigma_1$ and $\Sigma_2$ are the covariance matrices of the two classes with the off-diagonal entries set to zero. We set these terms to zero (as opposed to regularization) because we found that the off-diagonal terms were very poorly estimated for our data set. Calculating FLD weights in this manner is thus equivalent to weighting each unit by its d' alone (while ignoring any optimization that considers correlated activity between units).

In the case of the regular FLD (e.g. *Figure 6*), units were randomly selected for populations smaller than the full population size recorded. In the case of the ranked-FLD (*Figures 8c* and *9b*), weights were computed for each unit as described by *Equation 2* and then ranked by sign (such that positive weights were ranked higher than negative weights), and the top N units with the largest magnitude weights were selected for different population size N.

## Spike count classifier variants (SCC, ranked SCC)

For the SCC, the weight applied to each neuron was 1/N where N corresponded to the population size under consideration. The criterion was then computed as described above for the FLD. In the case of the regular SCC, units were randomly selected for populations smaller than the full population size recorded. In the case of the ranked-SCC (*Figure 9d*), weights were computed for each unit and ranked as described for the ranked FLD, and the top N units with the largest magnitude weights were selected for different population size N.

## Cross-validated training and testing

We applied the same, iterative cross-validated linear decoding procedure for each decoder. On each iteration of the resampling procedure, the responses for each unit were randomly shuffled within the set of matched n-back to ensure that artificial correlations (e.g. between the neurons recorded in different sessions) were removed. Each iteration also involved setting aside the responses to two randomly selected images at each n-back (presented as both novel and familiar, for four trials in total) for testing classifier performance. The remaining trials were used to train one of the four linear decoders to distinguish novel versus familiar images, where the novel and familiar classes included the data corresponding to all n-backs and all trial outcomes. Memory strength was measured as the dot product of the test data vectors $x$ and the weights $\mathbf{w}$, adjusted by $b$ (*Equation 1*). Histograms of these distributions for the FLD decoder are shown in *Figure 6b* across 1000 resampling iterations. A neural prediction of the proportion of trials on which 'familiar' would be reported was computed as the proportion of each distribution that took on a value less than the criterion (*Figure 6c*). This process was repeated for a broad range of population sizes and for each size, the mean squared error between the actual and predicted forgetting functions were computed to determine the best sized population (e.g. *Figure 6g*).

To compute predictions of reaction times on correct and error trials, we began by considering the proportion of the distributions shown in *Figure 6b* predicted to be reported 'correct' versus 'wrong', as a function of n-back, for both novel and familiar presentations (*Figure 6d*). Examination of these proportions plotted against the monkeys' reaction times that they map onto revealed a linear relationship (*Figure 6e*), which we fit with a line by minimizing mean squared error. The final neural predictions for reaction times were produced by passing the predicted proportions for correct and error trials through the resulting linear equation (*Figure 6f*).

## Estimating performance for larger-sized populations

To estimate performance for larger sized populations than those we recorded, we computed quantified how the mean and standard deviation of the distributions depicted in *Figure 6b*, as well as the value of the criterion, grew as a function of population size (*Figure 6—figure supplement 1*). For both the SCC and FLD, the trajectories of the means and the criterion were highly linear as a function of population size (*Figure 6—figure supplement 1a–b*, left), whereas the standard deviations plateaued (*Figure 6—figure supplement 1a–b*, right). We modeled the population response

distributions at each n-back (*Figure 6b*) as Gaussian, and we estimated the means and standard deviations of each distribution at different population sizes by extending the trajectories computed from our data to estimates at larger population sizes (*Figure 6—figure supplement 1* dotted lines). This process was similar in spirit but differed in detail for each decoder.

In the case of the SCC, the mean population response was computed as the grand mean spike count across the population, and consequently did not grow with population size (*Figure 6—figure supplement 1a*, left). We extended these trajectories with a simple linear fit to the values computed from the data. In contrast, the trajectory corresponding to standard deviation decreased as a function of population size (*Figure 6—figure supplement 1a*, right) and to extend these trajectories, we fit a two-parameter function:

$$SCC\_sd(x) = \left( \sum_{1}^{x} a^b \right)^{1/b} \qquad (5)$$

where x corresponds to population size and the parameters a and b were fit to the data.

In the case of the FLD, the population mean was computed as a weighted sum and grew linearly with population size (*Figure 6—figure supplement 1b*, left). We extended these trajectories with a linear fit to the values computed from the data. In contrast, the trajectories corresponding to the population standard deviations for each n-back grew in a nonlinear manner (*Figure 6—figure supplement 1b*, right), and we extend them by fitting the 2-parameter function:

$$FLD\_sd(x) = (ax)^b \qquad (6)$$

where x corresponds to population size and the parameters a and b were fit to the data.

For both the SCC and FLD decoders and their threshold variants, we computed behavioral predictions for larger sized populations by replacing the histograms in *Figure 6b* with Gaussians matched for the means and standard deviations determined by the extended trajectories, relative to the extended estimate for the criterion.

## Prediction quality:

To measure the prediction quality of the neural predictions for both the forgetting function and reaction time patterns, we developed a measure that benchmarked the MSE between the behavioral patterns and neural predictions by the worst-possible fit given that our procedure involves a global alignment of behavioral and neural data (*Figure 6g*). The worst-possible fit was computed as a step function, under the assumptions that performance as a function of n-back should be continuous, have non-positive slope, and be centered around chance. For example, the average proportion correct for the monkey's pooled behavioral forgetting function (*Figure 6g*) was 84%, and the benchmark was thus assigned as 84% proportion chose familiar for every n-back, and 16% for the novel images. Prediction quality was computed as:

$$PQ = 100 * \frac{MSE_{neural} - MSE_{benchmark}}{MSE_{neural}} \qquad (7)$$

where $MSE_{neural}$ and $MSE_{benchmark}$ correspond to the MSE between the actual behavioral forgetting function and the neural prediction or the benchmark, respectively.

To produce prediction quality estimates for reaction times (*Figure 6f*), the benchmark forgetting function was passed through the same procedure as the neural prediction to produce benchmarked reaction time predictions (*Figure 6f*, dotted). PQ was then computed as described in *Equation 7*.

## Simulation to estimate the impact of visual selectivity on population size

To estimate the impact of visual selectivity on population size (*Figure 9c*), we compared FLD and ranked-FLD behavioral prediction error trajectories (as a function of population size) for two simulated versions of our data: one that 'replicated' each unit and another that corresponded to 'visual modulation removed' (*Figure 9*). For these simulations, the strength of the visual memory signal for each unit was measured at each n-back as the mean proportional change in the spike count response for the same images presented as novel versus as familiar across all image pairs, and visual

memory modulation was modeled as multiplicative. In the case of the 'replicated' simulation, the novel and familiar responses to each image were determined by considering the average response to that image when it was novel versus familiar, and adjusting that quantity based on the proportional decrement computed for each n-back. For example, if the proportional decrement at 1-back for a unit was 10% and the unit responded to one image with an average (across the novel/familiar presentations) of 6 spikes, the replicated prediction for the novel and familiar presentation would be 6.32 spikes and 5.69 spikes, respectively (for a total difference of 0.63 spikes). If the same unit responded to a different image at 1-back with an average of 3 spikes, the replicated prediction would be 3.16 spikes and 2.84 spikes for novel and familiar images, respectively (for a total difference of 0.32 spikes). The process was repeated for each image by applying the proportional decrement determined for the n-back at which it was presented. These predictions were then converted into spike counts by applying Poisson trial variability. As a verification that this simulation captured the relevant aspects of the data, we compared its FLD behavioral prediction error trajectory to the error trajectory of the intact data (*Figure 9c*, gray versus black).

In the case of the 'visual modulation removed' simulation, the process was similar but instead of considering the actual response of the unit to a particular image, visual memory modulation was applied to the grand mean spike count across all images for that unit. A response prediction for each image was determined by applying the proportional decrement determined for the n-back at which it was presented around the grand mean spike count. These predictions were then converted into spike counts by applying Poisson trial variability.

### Unit d'

Unit d' was calculated, for each unit, as the difference in the mean responses to the set of images presented as novel versus the set presented as familiar, divided by the average standard deviation across the two sets (*Figure 8a–b*).

### Bootstrap statistical testing

To determine the fraction of units that produced responses that differed between novel versus familiar images or between the pre-stimulus and stimulus-evoked period, we computed p-values to evaluate the statistical significance of the observed differences in the mean values via a bootstrap procedure. On each iteration of the bootstrap, we randomly sampled the true values from each population, with replacement, and we computed the difference between the means of the two newly created populations. We computed the *p* value as the fraction of 1000 iterations on which the difference was flipped in sign relative to the actual difference between the means of the full data set (*Efron and Tibshirani, 1998*).

## Acknowledgements

We thank Alex Smolyanskaya for her contributions to early phases of this work. This work was supported by the National Eye Institute of the National Institutes of Health (award R01EY020851), the Simons Foundation (through an award from the Simons Collaboration on the Global Brain), and the McKnight Endowment for Neuroscience.

## Additional information

#### Competing interests

Nicole C Rust: Reviewing editor, *eLife*. The other author declares that no competing interests exist.

#### Funding

| Funder | Grant reference number | Author |
| --- | --- | --- |
| National Eye Institute | R01EY020851 | Travis Meyer Nicole Rust |
| Simons Foundation | Simons Collaboration on the Global Brain | Travis Meyer Nicole Rust |

| McNight Endowment for Neuroscience | Scholar Award to NCR | Nicole Rust |

The funders had no role in study design, data collection and interpretation, or the decision to submit the work for publication.

## Author contributions
Travis Meyer, Conceptualization, Investigation, Writing—original draft; Nicole C Rust, Conceptualization, Funding acquisition, Investigation, Writing—original draft

## Author ORCIDs
Travis Meyer (iD) https://orcid.org/0000-0003-4672-5368
Nicole C Rust (iD) https://orcid.org/0000-0002-7820-6696

## Ethics
Animal experimentation: All procedures were performed in accordance with the guidelines of the University of Pennsylvania Institutional Animal Care and Use Committee under protocol 804222.

## Decision letter and Author response
Decision letter https://doi.org/10.7554/eLife.32259.019
Author response https://doi.org/10.7554/eLife.32259.020

## Additional files

### Supplementary files
• Source data 1. Data used to compare neural predictions with behavioral responses. Behavioral data include the forgetting function and reaction times, each as a function of n-back. Neural data include the spike count responses of each unit to the same images presented as novel and as familiar at a range of n-back (Source_data.zip).
DOI: https://doi.org/10.7554/eLife.32259.016

• Transparent reporting form
DOI: https://doi.org/10.7554/eLife.32259.017

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
