## [Decision Letter]

Thank you for submitting your article "Single-exposure visual memory judgments are reflected in IT cortex" for consideration by *eLife*. Your article has been favorably evaluated by Timothy Behrens (Senior Editor) and three reviewers, one of whom is a member of our Board of Reviewing Editors. The following individual involved in review of your submission has agreed to reveal his identity: Guy A Orban (Reviewer #2).

The reviewers have discussed the reviews with one another and the Reviewing Editor has drafted this decision to help you prepare a revised submission.

Summary:

This manuscript investigates the neuronal signatures of image familiarity (or visual recall) decisions in inferotemporal cortex (IT) during a task in which monkeys indicated whether a presented image was novel (never seen before) or familiar (seen exactly once before in the same session). The authors optimized task difficulty by increasing the temporal delay between the first and second presentations (n-back) of the images. They demonstrate that monkeys can report whether a presented image is novel or familiar based only on one image viewing with delays as long as 64 intervening images (~about 5 minutes in their task paradigm). They characterize monkeys' behavior through forgetting functions, in which they show that the proportion of trials reported as familiar systematically decreases as a function of the delay between the first and second image presentations (n-back). They also analyze reaction time distributions as a function of n-back for correct and incorrect trials and find that reaction times increase with increasing delays for correct trials, but decrease with increasing delays for incorrect trials resulting in a x-shaped pattern. They ask whether IT neural signals can explain these features of monkeys' visual recall behavior. Specifically, they evaluate repetition suppression as a potential neuronal mechanism underlying this behavior – that is, can the difference in total number of spikes evoked by novel vs. familiar images across the population account for visual memory behavior? They argue that repetition suppression as applied equally to the entire IT population cannot be reconciled with their behavioral observations. The authors then put forward the hypothesis that a linear decoding mechanism which weights neurons by the amount of task-relevant information they carry is a neural decoder that can map IT neural signals to behavioral forgetting functions, reaction time patterns and recover monkey-wise differences. The authors suggest that a subset of IT neurons carry behaviorally relevant information about single-exposure visual memories and is governed by a learning rule that multiplicatively scales population vector representations in multi-dimensional perceptual space.

All reviewers were impressed by the quality of the study and the elegance of the design. The central question of how visual memories are stored and signaled is timely and interesting. The question is focused on behavior underlying single trials and the task to address that question is rigorously designed. This is the first study to rigorously test the repetition suppression hypothesis by combining well controlled single-shot familiarity behavior with IT electrophysiology. The computational and analytical approach employed by the authors in the context of visual familiarity is novel. Overall, this is an important study which is expected to be of high impact, and interest to a broad array of visual, systems, and computational neuroscientists. At the same time, we note several conceptual issues which could be addressed and clarified, as well as issues with data analysis that need to be resolved and taken into consideration in order to better explain experimental observations.

Essential revisions:

1) I'm uncertain how surprising the results are. The strength model that the authors propose seems a very plausible model for how the monkey decides old versus new, and *given this model*, the overall pattern of behavior results and reaction times would seem to follow almost necessarily, with the only question being how precise the fit is (since LDA assumes two normally distributed probability density functions). The analysis performed by the authors showing that using one monkey's neural data together with another monkey's behavior does a poorer job fitting the behavioral performance than using the same monkey's data and behavior speaks to this question, suggesting that the precise weights do matter. But for monkey 1 behavioral, monkey 2 neural, the difference doesn't seem all that big. And monkey 2 was the animal with greater repetition suppression. What happens when you scramble the identities of the neurons used by the classifier? How does the error between neural prediction and actual forgetting function change? The authors show that the weights are mostly positive, suggesting that perhaps this doesn't matter so much.

2) Related to point #1, what happens if you train using a different type of classifier (e.g. SVM). How do the weights from the two classifiers compare to each other? It seems the precise weights shouldn't matter, since one can add a vector in the null space of the projection vector and get the same classification performance. If you take a difference between the weights of a different classifier, and the Fisher LD, do you in fact get a vector in the null space of the Fisher LD? It would be good for the authors to jump outside their particular model, by comparing it to other models, so we have a better understanding of what is essential and what is circumstantial.

3) Visual memory is an example of one parameter that changes vector length. Any parameter that IT neurons are tolerant to – like size, shape, contrast – would also result in changes in vector length. How does this sort of a coding scheme reconcile with other parameters? Is familiarity or visual memory encoding different from these other parameters? For example, when you change the contrast of an object, would this produce an increase in response strength that would fool the classifier into thinking that an unfamiliar object is now familiar?

4) The paradigm was modified in a potentially important way between monkeys does this explain the behavioral difference between subjects? In monkey 1, since the targets were shown earlier, could this have caused attentional distraction, leading to worse performance.

5) Where do the visual images come from the authors used something like over 13k images; not so easy to find; the nature of the images might be a factor influencing familiarization.

6) The sign of the weights (Figure 6) could be tested statistically to support better the conclusions. Also, the analysis should be done for each monkey individually.

7) In Figure 7, we need some stat measure of match between data and model. For example, in monkey 2 the prediction seems outside the confidence interval of the actual data for 64-back.

8) The authors compare IT suppression signals as a candidate mechanism to account for the monkeys' behaviors. However, since IT suppression signals at 64-back are extremely small (even enhanced to a small extent) and cannot account for 64-back behavior (compare Figure 5 with 3B and C) in which monkeys do report that they viewed a familiar image on some fraction of the trials, the authors conclude that the number of spikes hypothesis is insufficient to explain behavior.

8a) Is it possible to re-parameterize the suppression index (shown in Figure 5) in the form of a forgetting function plotted against n-back (like in Figure 7 for the weighted linear classifier)? Right now, the author's argument against the suppression model is rather hand wavy, appealing only to the 64-back condition. A quantitative comparison using the same currency (forgetting function) would be useful.

8b) The crucial comparison of whether IT responses align with behavior should be made monkey-wise, since the behavior between the monkeys is quite variable at the point of interest (64-back). The suppression indices vs. n-back plot is provided (Figure 7) separately for both monkeys. However, examining these plots gives an impression that IT suppression indices for individual monkeys do predict individual monkeys' behavior. For example, comparing 7A (7B) and the green curve in 7C (7D), we see that for monkey 1, at 64-back, there is an enhancement (as opposed to suppression), this means that responses for the second viewing of images at 64-back leads to a higher response than the first viewing. This results in the monkey's behavior to choose novel on a larger fraction of these trials (80% proportion novel=20% proportion familiar). These observations make it difficult to estimate how much extra variance in behavior is explained by the weighted linear model, and questions the necessity to invoke this model if individual behavior can be explained by the monkeys' respective IT population read-outs?

8c) Perhaps even splitting the 64-back data into correct versus error trials and correlating neuronal activity for correct versus error trials might be a more straightforward way of implicating IT neuronal populations in visual memory behavior.

9a) The analysis to estimate the correct population size while comparing IT neural and behavioral responses produces a minimum mean squared error at population size N=800. This seems like a very large neuronal population is required for this behavior, and is an intriguing finding with interesting implications. This finding could manifest in at least two ways: i) IT neuronal populations are extremely selective, thus responding robustly only to a handful of images every day. Are individual neuronal rank curves for these images steep, responding only to a few preferred images and not responding to the rest? ii) Alternatively, even if IT neurons are sufficiently selective to the presented images, the reduction from first to second presentation is not present for all images for all neurons, the signal itself is not reproducible from the first to the second presentation in all neurons. It would be important to know which of these hypotheses explain your neuronal data? Would you be able to quantify the sources of variability/noise that result in requiring a large neuronal population? Overall, we need some better understanding of how the judgement about old/new relates to responses at the single cell level.

9b) Related to the previous point: The large number of neurons required to make the old/new judgment seems like it could be an artifact of how the authors created their pseudopopulation, since the neurons were actually responding to different images. It could be that if you just have 100 neurons' response to the same image, both old and new, then you can already do a great job classifying old/new for that image. Can the authors please comment on this?

10) When you increase the neuronal population with simulated units beyond N=800, why is there an increase in MSE error? What is the intuition for that? Additional discussion of this point would be helpful. The authors also need to explain more clearly how they extrapolated their populations to N=800+ in each monkey by simulated units. None of the reviewers could understand the explanation currently in the Materials and methods.

11) The neural data recorded from an animal is unable to predict performance for the same animal, since you do not have 800 neurons from one animal, but from both combined. What does this tell us about the capacity of IT populations to support visual memory behavior?

12) Predicting individual differences using a decoding model for N=2 is a statistically underpowered problem. When you add in the fact that one (out of two) of the monkeys (monkey 1) had a bias or performed worse at the task at higher delays makes it further problematic to make a compelling case for individual differences. Perhaps splitting your data for each monkey into two subsets based on session – high performance sessions vs. low performance sessions (based on some criterion on the forgetting functions) and consequently being able to predict high vs. low performance from IT neural data within the same monkey might be a more powerful test of the model.

13) The authors need to explain how they created the pseudopopulation in more detail, and they should do so in the main text, since this is a crucial point. For example, it's not clear from the Materials and methods how trials were chosen from each day to create the aligned pseudopopulations, since presumably different sessions had different numbers of correct and incorrect trials. They should clarify what they mean in the second paragraph of the subsection “A weighted linear read-out of IT accurately predicts behavior:”.

14) The reviewers found the Discussion somewhat lacking. Here are some suggestions for improvement:

• First paragraph “linear weights were largely positive”, very loose sentence with important consequences as it is the conclusion of the paper please provide more support, or tone down your claim.

• Second paragraph makes a link between type of coding; bell-shaped or monotonic and read-out; concludes that the present readout is compatible with bell shaped coding, while all recent studies (e.g. Sheinberg, Tsao, Vogels) suggest that IT neurons mainly use a monotonic coding. The authors should discuss this.

• Third paragraph about previous monkey experiments is a very weak paragraph simply re-quoting papers used in the Introduction; the authors missed a golden opportunity to compare their mechanism to others know in the monkey e.g. the short term single stimulus buffer involved in successive discriminations described by Vogels and Orban J Neurophysiol 1994 (see also Tsao Nat NS this year); the sites positively involved in familiar face processing described by Freiwald in his recent Science paper (2017); the memory mechanism in parietal cortex recently described by the Miyashita group. Some of these studies are briefly mentioned in the Introduction, but further discussion, in the context of the results found by the authors, would be useful.

• Fourth paragraph is better and should probably be moved to the end.

• Fifth paragraph about human work is superficial, and is missing the issue of susceptibility artefacts in FMRI of rostral temporal regions (see e.g. Georgieva et al., J NS 2009); That is why PET was use d for a long time for semantic studies involving temporal pole (now double echo sequences are used in fMRI); Hence a very relevant reference is an old PET study Vandenberghe et al. Neuroimage 1995 describing decreased responses in rostral inferior temporal cortex.

• Sixth paragraph is an almost literal copy of a paragraph from Results: should come first in the discussion of the methodology followed in the experiment.

• An interesting question that comes out of this study is whether (and how much) state-of-the-art deep convolutional neural networks of visual object recognition/processing can signal the percept of visual memories. How can current architectures be improved to support visual recall? In which layers do these computations reside? This might be an interesting point of discussion.

---

## [Author Response]

Essential revisions:1) I'm uncertain how surprising the results are. The strength model that the authors propose seems a very plausible model for how the monkey decides old versus new, and given this model, the overall pattern of behavior results and reaction times would seem to follow almost necessarily, with the only question being how precise the fit is (since LDA assumes two normally distributed probability density functions). The analysis performed by the authors showing that using one monkey's neural data together with another monkey's behavior does a poorer job fitting the behavioral performance than using the same monkey's data and behavior speaks to this question, suggesting that the precise weights do matter. But for monkey 1 behavioral, monkey 2 neural, the difference doesn't seem all that big. And monkey 2 was the animal with greater repetition suppression. What happens when you scramble the identities of the neurons used by the classifier? How does the error between neural prediction and actual forgetting function change? The authors show that the weights are mostly positive, suggesting that perhaps this doesn't matter so much.

We have incorporated a number of new analyses that address these questions, including a quantitative comparison of the spike count versus weighted linear classifier (Figure 7) and “topranked” variants of each of those classifiers (Figure 8). We have also incorporated a benchmarked metric for comparing the two classifiers (prediction quality, ‘PQ’) and a new way to visualize how the spike count decoding scheme fails (Figure 7). Our interpretation of these results is presented in the last paragraph of the subsection “The single-unit correlates of the weighted linear decoding scheme”. While one could argue that the FLD should have higher neural population performance in the limit of large population size (e.g. higher neural d’), it is not obvious to us that it should be a better predictor of behavior when overall population performance is matched. We have also determined why that is the case – it is true that within the subpopulation of 50% top ranked units, the specific pattern of weights does not matter when determining behavioral predictions insofar as counting spikes within the 50% top-ranked units is nearly as effective as the FLD (Figure 8). However, something has to be done about the bottom 50% ranked d’ units (either the application of FLD weights or thresholding them away to weights of 0) to get the most accurate neural predictions of behavioral performance.

2) Related to point #1, what happens if you train using a different type of classifier (e.g. SVM). How do the weights from the two classifiers compare to each other? It seems the precise weights shouldn't matter, since one can add a vector in the null space of the projection vector and get the same classification performance. If you take a difference between the weights of a different classifier, and the Fisher LD, do you in fact get a vector in the null space of the Fisher LD? It would be good for the authors to jump outside their particular model, by comparing it to other models, so we have a better understanding of what is essential and what is circumstantial.

To address these issues, we have added 3 new types of classifiers in our revision (SCC, ranked-SCC, and ranked-FLD; Figure 7–Figure 8). Our conclusions are described above for point 1. As a minor point: we did not incorporate the SVM into our manuscript because it performs less well than the FLD (consistent with our past experience with other data sets, it overfits this dataset).

3) Visual memory is an example of one parameter that changes vector length. Any parameter that IT neurons are tolerant to – like size, shape, contrast – would also result in changes in vector length. How does this sort of a coding scheme reconcile with other parameters? Is familiarity or visual memory encoding different from these other parameters? For example, when you change the contrast of an object, would this produce an increase in response strength that would fool the classifier into thinking that an unfamiliar object is now familiar?

We have added this issue to the Discussion (fifth paragraph). Because our images were randomly selected, we do not have the power to address these questions in our current data but we plan to address them in a near future study.

4) The paradigm was modified in a potentially important way between monkeys does this explain the behavioral difference between subjects? In monkey 1, since the targets were shown earlier, could this have caused attentional distraction, leading to worse performance.

Possibly. We have included this point in the Discussion (third paragraph).

5) Where do the visual images come from the authors used something like over 13k images; not so easy to find; the nature of the images might be a factor influencing familiarization.

We have added this information to our Materials and methods section (subsection “The single-exposure visual memory task”, third paragraph) and Figure 2—figure supplement 1.

6) The sign of the weights (Figure 6) could be tested statistically to support better the conclusions. Also, the analysis should be done for each monkey individually.

Histograms for each monkey are now included as Figure 8 and the requested statistics are presented in the Results (subsection “The single-unit correlates of the weighted linear decoding scheme”, second paragraph).

7) In Figure 7, we need some stat measure of match between data and model. For example, in monkey 2 the prediction seems outside the confidence interval of the actual data for 64-back.

Our revision incorporates the measure “prediction quality” (PQ), which benchmarks mean square error between the best-case and worst-case fits. This measure is described in– the last paragraph of the subsection “Predicting behavioral response patterns from neural signals”.

8) The authors compare IT suppression signals as a candidate mechanism to account for the monkeys' behaviors. However, since IT suppression signals at 64-back are extremely small (even enhanced to a small extent) and cannot account for 64-back behavior (compare Figure 5 with 3B and C) in which monkeys do report that they viewed a familiar image on some fraction of the trials, the authors conclude that the number of spikes hypothesis is insufficient to explain behavior.8a) Is it possible to re-parameterize the suppression index (shown in Figure 5) in the form of a forgetting function plotted against n-back (like in Figure 7 for the weighted linear classifier)? Right now, the author's argument against the suppression model is rather hand wavy, appealing only to the 64-back condition. A quantitative comparison using the same currency (forgetting function) would be useful.

To address this issue, we have introduced a new “spike count classifier” and pitted it against the FLD using comparable procedures (Figure 7); the FLD is in fact a better predictor of behavior in both monkeys. We have also introduced a new visualization of the alignment of the neural predictions and behavioral data across all n-back (Figure 7) to get a better sense of why the spike count decoding scheme fails.

8b) The crucial comparison of whether IT responses align with behavior should be made monkey-wise, since the behavior between the monkeys is quite variable at the point of interest (64-back). The suppression indices vs. n-back plot is provided (Figure 7) separately for both monkeys. However, examining these plots gives an impression that IT suppression indices for individual monkeys do predict individual monkeys' behavior. For example, comparing 7A (7B) and the green curve in 7C (7D), we see that for monkey 1, at 64-back, there is an enhancement (as opposed to suppression), this means that responses for the second viewing of images at 64-back leads to a higher response than the first viewing. This results in the monkey's behavior to choose novel on a larger fraction of these trials (80% proportion novel=20% proportion familiar). These observations make it difficult to estimate how much extra variance in behavior is explained by the weighted linear model, and questions the necessity to invoke this model if individual behavior can be explained by the monkeys' respective IT population read-outs?

These comparisons are now presented monkey-wise (Figure 7) along with prediction quality measures that quantify the degree to which the FLD is a better behavioral predictor than the SCC (Figure 7).

8c) Perhaps even splitting the 64-back data into correct versus error trials and correlating neuronal activity for correct versus error trials might be a more straightforward way of implicating IT neuronal populations in visual memory behavior.

Thank you for the suggestion. We tried this, both for the pseudopopulation and for the data recorded (simultaneously across units) in each session. Unfortunately, our data is statistically underpowered to address this question.

9a) The analysis to estimate the correct population size while comparing IT neural and behavioral responses produces a minimum mean squared error at population size N=800. This seems like a very large neuronal population is required for this behavior, and is an intriguing finding with interesting implications. This finding could manifest in at least two ways: i) IT neuronal populations are extremely selective, thus responding robustly only to a handful of images every day. Are individual neuronal rank curves for these images steep, responding only to a few preferred images and not responding to the rest? ii) Alternatively, even if IT neurons are sufficiently selective to the presented images, the reduction from first to second presentation is not present for all images for all neurons, the signal itself is not reproducible from the first to the second presentation in all neurons. It would be important to know which of these hypotheses explain your neuronal data? Would you be able to quantify the sources of variability/noise that result in requiring a large neuronal population? Overall, we need some better understanding of how the judgement about old/new relates to responses at the single cell level.

We very much appreciate the concreteness and clarity of this suggestion. To address these issues, we have introduced new sections to the paper (subsections “The single-unit correlates of the weighted linear decoding scheme” and “The impact of visual selectivity on population size”) as well as two new Figures (Figure 8–Figure 9). The take home message (presented in the last paragraph of the aforementioned subsection) is that the large population size requirement follows from a combination of visual memory signals that combined across only the subset of ~50% IT units, as well as from visual selectivity.

9b) Related to the previous point: The large number of neurons required to make the old/new judgment seems like it could be an artifact of how the authors created their pseudopopulation, since the neurons were actually responding to different images. It could be that if you just have 100 neurons' response to the same image, both old and new, then you can already do a great job classifying old/new for that image. Can the authors please comment on this?

We have addressed the question of ‘How many responsive neurons would you need to account for behavior?’ in Figure 9. We have also elaborated the description about how our pseudopopulation was created (subsection “Predicting behavioral response patterns from neural signals”, second paragraph). Under the assumption that the data we record in any one session are a representative sample of the responses across the IT population, our methods should not produce over-estimates; we have incorporated this assumption into the description.

10) When you increase the neuronal population with simulated units beyond N=800, why is there an increase in MSE error? What is the intuition for that? Additional discussion of this point would be helpful. The authors also need to explain more clearly how they extrapolated their populations to N=800+ in each monkey by simulated units. None of the reviewers could understand the explanation currently in the Materials and methods.

We have addressed the question about the increase in MSE with population size in the fifth paragraph of the subsection “Predicting behavioral response patterns from neural signals” and with the addition of Figure 6. We have added Figure 6—figure supplement 1 to better explain how the populations were extrapolated to larger sizes in simulation.

11) The neural data recorded from an animal is unable to predict performance for the same animal, since you do not have 800 neurons from one animal, but from both combined. What does this tell us about the capacity of IT populations to support visual memory behavior?

We were a bit uncertain about what was being asked here. Our interpretation, rephrased is: “Clarify how you can predict each individual animal’s behavior given that doing so requires ~800 units and you have only recorded 400.” Our revision now includes a better explanation of how we extrapolate beyond the data that we have recorded to larger population sizes (Figure 6—figure supplement 2). Our ability to do this relies on the fact that the properties of a population’s units set the parameters that determine how population performance at each nback will grow along well-defined trajectories as a function of population size, and we can thus extend those trajectories to estimate how the same population would perform if more units were sampled.

12) Predicting individual differences using a decoding model for N=2 is a statistically underpowered problem. When you add in the fact that one (out of two) of the monkeys (monkey 1) had a bias or performed worse at the task at higher delays makes it further problematic to make a compelling case for individual differences. Perhaps splitting your data for each monkey into two subsets based on session – high performance sessions vs. low performance sessions (based on some criterion on the forgetting functions) and consequently being able to predict high vs. low performance from IT neural data within the same monkey might be a more powerful test of the model.

We have directly acknowledged the concerns about not having the power to determine individual differences (Discussion, first paragraph) and we have backed off claims about individual differences throughout the manuscript. We do continue to present the “same monkey vs. cross monkey” analysis (Figure 10) but with reduced and redirected emphasis. We appreciate the suggestion of splitting the data by performance across sessions within each monkey and we tried it, however, we have determined that we do not have sufficient dispersion across sessions to perform this analysis.

13) The authors need to explain how they created the pseudopopulation in more detail, and they should do so in the main text, since this is a crucial point. For example, it's not clear from the Materials and methods how trials were chosen from each day to create the aligned pseudopopulations, since presumably different sessions had different numbers of correct and incorrect trials. They should clarify what they mean in the second paragraph of the subsection “A weighted linear read-out of IT accurately predicts behavior”.

We have elaborated this section considerably (subsection “Predicting behavioral response patterns from neural signals”, second paragraph).

14) The reviewers found the Discussion somewhat lacking. Here are some suggestions for improvement:

We appreciate the feedback.

• First paragraph “linear weights were largely positive”, very loose sentence with important consequences as it is the conclusion of the paper please provide more support, or tone down your claim.

We have edited this text (Discussion, first paragraph) to reflect the additional analyses that have been incorporated into the revision to support these claims.

• Second paragraph makes a link between type of coding; bell-shaped or monotonic and read-out; concludes that the present readout is compatible with bell shaped coding, while all recent studies (e.g. Sheinberg, Tsao, Vogels) suggest that IT neurons mainly use a monotonic coding. The authors should discuss this.

We now see that this paragraph was misguided and confusing in the context of these other studies and so we have removed it. In its place, we have included a discussion of how the brain might disambiguate stimulus variables known to change total spike numbers (like image contrast, luminance and identity) from memory under this decoding scheme (Discussion, fifth paragraph), as well as the Chang/Tsao reference for monotonic coding.

• Third paragraph about previous monkey experiments is a very weak paragraph simply re-quoting papers used in the Introduction; the authors missed a golden opportunity to compare their mechanism to others know in the monkey e.g. the short term single stimulus buffer involved in successive discriminations described by Vogels and Orban J Neurophysiol 1994 (see also Tsao Nat NS this year); the sites positively involved in familiar face processing described by Freiwald in his recent Science paper (2017); the memory mechanism in parietal cortex recently described by the Miyashita group. Some of these studies are briefly mentioned in the Introduction, but further discussion, in the context of the results found by the authors, would be useful.

We have revised this paragraph as suggested (Discussion, second paragraph).

• Fourth paragraph is better and should probably be moved to the end.

We have moved it.

• Fifth paragraph about human work is superficial, and is missing the issue of susceptibility artefacts in FMRI of rostral temporal regions (see e.g. Georgieva et al., J NS 2009); That is why PET was use d for a long time for semantic studies involving temporal pole (now double echo sequences are used in fMRI); Hence a very relevant reference is an old PET study Vandenberghe et al. Neuroimage 1995 describing decreased responses in rostral inferior temporal cortex.

We have removed some superficial sentences and incorporated the Vanderberghe reference (Discussion, fourth paragraph).

• Sixth paragraph is an almost literal copy of a paragraph from Results: should come first in the discussion of the methodology followed in the experiment.

This paragraph has been removed.

• An interesting question that comes out of this study is whether (and how much) state-of-the-art deep convolutional neural networks of visual object recognition/processing can signal the percept of visual memories. How can current architectures be improved to support visual recall? In which layers do these computations reside? This might be an interesting point of discussion.

This point has been inserted into the Discussion (last paragraph).